# Behavioural individuality in clonal fish arises despite near-identical rearing conditions

David Bierbach[1,*], Kate L. Laskowski[1,*] & Max Wolf[1]

Behavioural individuality is thought to be caused by differences in genes and/or environmental conditions. Therefore, if these sources of variation are removed, individuals are predicted to develop similar phenotypes lacking repeatable individual variation. Moreover, even among genetically identical individuals, direct social interactions are predicted to be a powerful factor shaping the development of individuality. We use tightly controlled ontogenetic experiments with clonal fish, the Amazon molly (*Poecilia formosa*), to test whether near-identical rearing conditions and lack of social contact dampen individuality. In sharp contrast to our predictions, we find that (i) substantial individual variation in behaviour emerges among genetically identical individuals isolated directly after birth into highly standardized environments and (ii) increasing levels of social experience during ontogeny do not affect levels of individual behavioural variation. In contrast to the current research paradigm, which focuses on genes and/or environmental drivers, our findings suggest that individuality might be an inevitable and potentially unpredictable outcome of development.

[1] Department of Biology and Ecology of Fishes, Leibniz Institute of Freshwater Ecology and Inland Fisheries, Müggelseedamm 310, 12587 Berlin, Germany. * These authors contributed equally to this work. Correspondence and requests for materials should be addressed to D.B. (email: david.bierbach@gmx.de) or to K.L.L. (email: kate.laskowski@gmail.com).

Behavioural individuality—that is, repeatable and predictable among-individual differences in behaviour—is a ubiquitous phenomenon in humans[1,2] and a wide range of other animal species[3–8]. But what causes us and individuals within other animal species to be different from each other? Across the behavioural sciences, a common view is that individuality is caused by between-individual differences in genes and environmental conditions, including the social environment. One powerful way to test this paradigm is to investigate genetically identical individuals reared in isolation under 'identical' (that is, highly standardized) environmental conditions: if genetic and environmental differences are the key drivers of individuality, individuality should be largely absent (that is, not develop) in such a setting; moreover, the contribution of any other explanatory factor (for example, social rearing conditions) can be best evaluated against this baseline scenario.

Previous research has shown that substantial between-individual variation in morphological and physiological traits still develops even among genetically identical individuals reared under highly standardized conditions[9–11]. This work suggests that—even in the absence of genetic and environmental differences—maternal and epigenetic effects and/or minute experiential/environmental differences during development act as important drivers underlying phenotypic variation[12–14]. Up to now, only a handful of studies have investigated whether these same mechanisms can promote behavioural individuality in the absence of genetic and environmental differences between individuals[10,15–18]. Most prominently, recent studies on highly inbred mice find that behavioural individuality emerges among genetically identical individuals[19,20] when reared in the same environment. The interpretation of these findings, however, is hampered by the fact that individuals were reared in groups. Direct social interactions among conspecifics are well known to be a powerful factor affecting the development of individuality. Depending on the scenario, direct social interactions may either promote the development of individuality (for example, via the formation of social interaction structures like social hierarchies; via processes like frequency dependence, niche and role specialization[7,21–25]) or inhibit the development of individuality (for example, via positive frequency-dependent social learning and benefits of conformity[26–28]).

At present, it is thus not clear (i) whether and to what extent individuality emerges among genetically identical individuals reared in isolation under highly standardized environmental conditions and (ii) how a change in social rearing conditions affects the development of individuality. The goal of this study was to investigate these two central issues about the causes of behavioural individuality. To test this, we split broods of a genetically identical clonal fish, the Amazon molly (*Poecilia formosa*)[29,30], among three treatments that differed in the opportunity for social interactions (Fig. 1) from no social experience at all (0-day treatment), to a moderate level of social experience (7-day treatment) and finally extensive social experience (28-day treatment). Social interactions are of importance in this species: this species occurs in large shoals in nature[30] and previous work has shown early social experiences can have long-lasting impacts on later adult behaviour[31]. We then repeatedly measured each individual's behaviour at 7 weeks of age to assess levels of individuality. The 0-day treatment fish lack variation in the factors (genes and environment) that are currently thought to generate individual variation in behaviour, thus providing a baseline for the level of variation we should expect under the most controlled conditions. We can then investigate whether and how the addition of other factors, such as social rearing conditions, significantly changes the pattern of individual behavioural variation compared to this baseline scenario. Our results show that significant repeatable individual variation, that is, individuality, emerges even in isolated fish and that the increasing opportunity for social interactions in the other treatments does not change this pattern. These findings suggest that there may be further causes to the development of behavioural individuality than solely variation in genes or environment.

## Results

**Individuality emerges in near-identical animals**. In contrast to the prediction that individuality should not persist among genetically identical individuals reared under highly standardized environmental conditions, we found that substantial repeatable individual behavioural differences, that is individuality, developed in our 0-day treatment (Table 1 and Fig. 2). Overall, among-individual differences (that is, repeatability) accounted for roughly 30% of the total behavioural variation (Table 1 and Fig. 2).

**Social experience does not strengthen individuality**. Moreover, we found no evidence that increasing levels of social experience influenced the magnitude of behavioural individuality compared

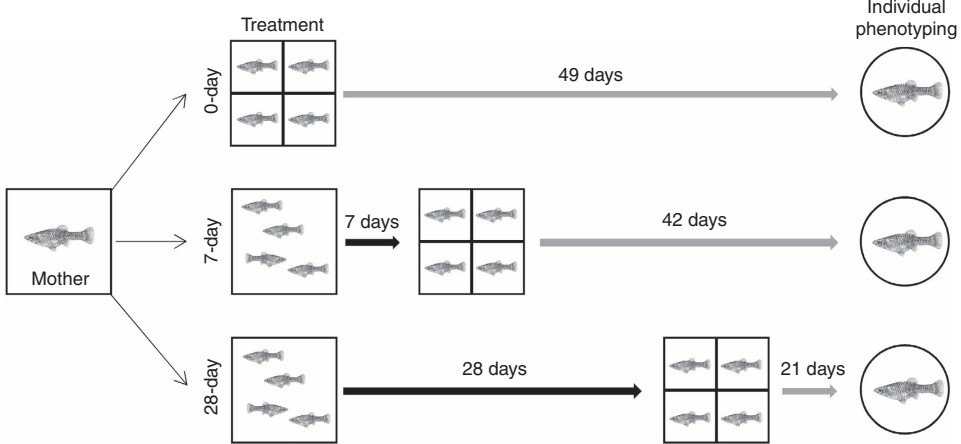

**Figure 1 | Schematic of experimental design.** In the 0-day treatment, directly after birth, genetically identical individuals were isolated (grey arrows) and housed in the same tightly regulated environmental conditions. In the 7-day and the 28-day treatments, genetically identical individuals were reared with increasing durations of social experiences (black arrows) and were then isolated (grey arrows) in the same environments. In all treatments, we assessed the behavioural phenotype of the individuals after 7 weeks.

**Table 1 | Result from global linear mixed model for activity in an open-field test.**

| Fixed effects | | Estimate (95%CIs) | |
|---|---|---|---|
| Standard length | | 0.12 (−0.37, 0.61) | |
| Observation | | 2.63 (2.11, 3.13) | |
| | | | |
| *Treatment* | | | |
| 0-day | | 7.79 (−4.00, 18.72) | |
| 7-day | | 7.41 (−4.58, 17.73) | |
| 28-day | | 9.08 (−2.17, 20.60) | |
| | | | |
| **Variance estimates** | | | |
| Among-mother | | 3.89 | |
| | **0-day** | **7-day** | **28-day** |
| Among-individual variance | 19.36 (5.28, 36.28) | 17.99 (3.34, 34.83) | 11.35 (1.59, 22.61) |
| Within-individual variance | 39.35 (28.37, 51.39) | 30.88 (22.23, 39.98) | 26.21 (19.11, 34.36) |
| Repeatability | 0.35 (0.13, 0.52) | 0.35 (0.13, 0.57) | 0.29 (0.09, 0.51) |

Our model with (square-root transformed) total distance swam as dependent variable included body size ('standard length') and observation (trial 1–4) as covariates and 'treatment' as fixed factor. Total behavioural variation was partitioned into its component parts: among-mothers across treatments ('among-mother variance') and among-individuals and within-individuals (that is, residual) within treatments. Repeatability estimates the proportion of the total variance that is due to among-individual differences within a treatment. Values in parentheses indicate the 95% credibility intervals; values not overlapping zero for fixed effects indicate that the estimate is significantly different from zero; CI's for variance estimates are constrained to be positive, therefore, we tested for the support of variance estimates by comparing the DIC of models with and without the random effect (see main text).

to the baseline 0-day treatment (Table 1 and Fig. 2). In our 7-day and 28-day treatments, as in our 0-day treatment, we observed the emergence of substantial among-individual differences in behaviour that are repeatable (Table 1 and Fig. 2). Neither the level of among-individual variation, nor the amount of total behavioural variation differed between our three treatments (Table 1). Indeed, the model containing treatment-specific variance estimates was not better supported than a model where individual variance was constrained to be the same across all three treatments ($\Delta$DIC = +0.667), indicating that levels of repeatable among-individual variation are similar regardless of social experience. Additionally, the inclusion of mother identity as a random effect was not well supported ($\Delta$DIC = −0.445) and including individual random slopes did not improve model fit (Supplementary Table 1). If anything, individuals in the treatments with the most social experience tended to exhibit lower among-individual and within-individual variation (Table 1). Furthermore, there were no differences in overall activity levels (treatment estimates in Table 1) or in average body size (0-day: 22.89 mm, 95% confidence interval: (21.75, 24.12); 7-day: 22.34 mm (21.11, 23.51); 28-day: 22.70 mm (21.52, 23.90)). We note that the absolute levels of behavioural variation exhibited by these clonal individuals closely resemble that which we have seen in non-clonal fish measured in a similar way (in the Atlantic molly, *Poecilia mexicana*, one of the parental species of *P. formosa*)[32].

## Discussion

Here we report experimental evidence that substantial behavioural individuality emerges even among genetically identical individuals housed under nearly 'identical' (that is, highly standardized) environmental conditions. This finding is in contrast to the current research paradigm associated with adaptive individuality, which focuses on differences in genes and/or environmental conditions (including the social environment) as drivers of individual behavioural variation. Importantly, our findings suggest that other, yet unidentified factors, must contribute to the establishment of individuality.

First, one potential factor that might have a stronger contribution on individuality than previously thought is minute and stochastic experiential/environmental variation between

individuals. In nature, no two individuals experience identical environmental conditions over development; similarly, experimentally, it is practically impossible to provide identical conditions to different individuals. Thus, despite our best efforts, it is likely that different individuals experienced different microenvironments such as slight differences in water temperature, olfactory signals or distribution of prey items. Recent theory on the adaptive development of individuality suggests that even among initially identical individuals, such minute environmental or experiential differences can induce positive feedback loops that eventually fix individuals on different developmental trajectories[33–36]. As Amazon mollies are known to use olfactory cues to detect conspecifics[37] and lineage-kin[38], it is conceivable that stochastic variation in such chemical cues between individuals (in combination with positive feedbacks) may be a factor driving the development of individuality in our experiments. Thus, in the present study, minute differences in chemical cues experienced by individuals across treatments may have contributed to distinct developmental trajectories. Further, the fish in our study received two types of food (see Methods section), which may have also contributed as a source of stochastic variation. The importance of such seemingly minute environmental differences are not yet well incorporated in the theoretical literature but in combination with potential feedback mechanisms could provide one potential mechanism for the development of individuality in otherwise 'identical' animals. Future work that closely follows the developmental experiences and the associated behavioural responses of individuals should aim to elucidate when and how such minute experiential differences can trigger the development of individuality. Moreover, an especially compelling question is whether developmental divergence triggered by such minute experiential differences makes the emergence and patterning of individuality inherently unpredictable.

A second potential driver of individuality, for which we could not experimentally control, may be epigenetic variation among individuals, which may be either stochastic or environmentally induced. Whether and under which conditions such between-individual epigenetic variation (and thus the observed individuality) is the result of an adaptive strategy is currently an open question. Mothers may, for example, employ a bet-hedging strategy to generate adaptive epigenetic variation among her offspring[12,14]; this may be particularly beneficial in unpredictably

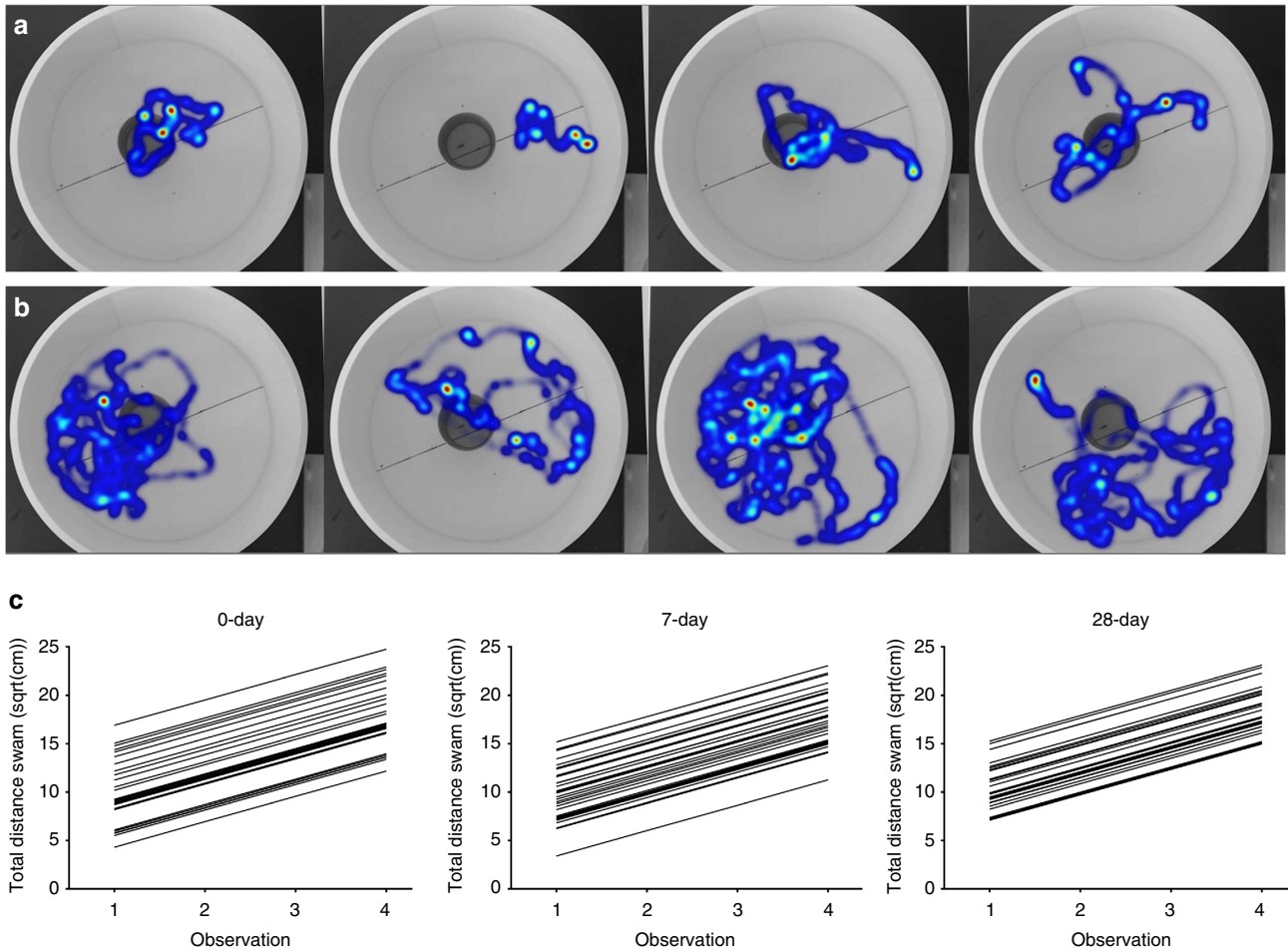

**Figure 2 | Individual behaviour in an open-field test over four repeated observations.** (**a**,**b**) show examples of a less active and a highly active individual, respectively; shown are heat map outputs produced by EthoVision software for each of the four trials. (**c**) Individual activity levels for the three treatments are shown. Each line represents one individual with the predicted intercept and slope from the models ($N = 31$ for 0-day and 7-day treatments and $N = 32$ for 28-day treatment). In all three treatments, we observe substantial among-individual differences in behaviour that are repeatable. Neither the level of among-individual variation, nor the amount of total behavioural variation differed between our three treatments (Table 1).

changing environments or in clonal organisms where standing genetic variation is lacking. Furthermore, between-individual epigenetic variation may be an adaptive response to minute experiential/environmental differences between individuals (see above), thereby potentially contributing to a positive feedback loop reinforcing such differences[11,14]. Alternatively, variation among individuals may not be adaptive at all, but reflect stochasticity or noise during development which is then canalized as a result of the developmental process[11,39–43].

It is currently thought that direct social interactions are a powerful causal factor affecting the development of individuality. As discussed above, dependent on the species ecology and other factors such as density, risk and so on, direct social interactions can be predicted to either promote (for example, via the formation of social interaction structures like social hierarchies; via processes like frequency dependence, niche and role specialization[7,21–25]) or inhibit (for example, via positive frequency-dependent social learning and benefits of conformity[26–28]) the development of individuality. Up to now, however, few studies[16] have evaluated the importance of direct social interactions with a controlled experimental approach that compares a treatment in which genetically identical individuals are allowed to directly interact with each other with an appropriate baseline treatment that does

not allow for such interactions. When comparing our 0-day baseline treatment with our 7-day and 28-day direct social experience treatments, we find that the amount of behavioural individuality observed is not affected by the level of direct social experience of individuals. We note that—while we do not find statistically significant differences between treatments—the amount of among- and within-individual variation does tend to decrease in the most social treatment (28-day). This finding is in line with previous studies on clonal fish, which did not find an effect of direct social experience on the repeatability of behaviour[16]. We do not claim that social processes play no role in the development of behavioural individuality, but our findings strongly suggest that other more nuanced factors may be substantially more important for the development of individuality than currently thought.

Over the last decades, substantial research effort has been spent investigating the causes of behavioural individuality and much of this research has aimed to explain and predict behavioural individuality with differences in genes and/or environmental conditions. Regardless of the exact causes of individuality in our experiments, our findings suggest that individuality may be a more general phenomenon and potentially an inevitable and inherently unpredictable outcome of the development of complex phenotypes.

## Methods

**Animal care and maintenance.** Stock populations of *P. formosa* (Amazon molly, obtained from Manfred Schartl, University of Würzburg) are maintained in large (100 l) stock aquariums. The all-female Amazon molly originates from a single natural hybridization event between the sailfin molly *Poecilia latipinna* and the Atlantic molly *Poecilia mexicana*[44,45]. It reproduces gynogenetically and females require sperm from one of the parental species to stimulate egg production[45–47]. Therefore, several (two–four) males of *P. mexicana* were kept with each stock population aquarium. Stock populations experience ambient light conditions similar to the local light cycle (~14:10 L:D). Fish were fed *ab libitum* three times daily on standard flake fish food. We performed weekly water changes to replace ~30% of the total water volume of each tank. To generate the experimental individuals, we isolated gravid females from a single isogenic line (strain 1304, lab code Manfred Schartl) in separate 35 l tanks containing a gravel bottom and plastic plants. To further limit the potential for differences among mothers (for example, in maternal effects), these gravid females were all sisters from the same brood ensuring all individuals were the same age, from the same mother and had experienced similar conditions for their entire lives. Intermittent genetic samplings of this strain confirm that all individuals are clones. We checked females' tanks twice daily for evidence of offspring. In case a female gave birth, we took 12 juveniles from the brood and randomly assigned four siblings to each of our three treatments (see more details below; Fig. 1). This split-brood design helped to ensure that any differences in maternal effects were at least split among all our treatments. Additionally, we only used broods of similar size (12–24 offspring) to reduce the potential for differences in maternal provisioning. In total, seven different mothers (one mother contributed two clutches) contributed to the experimental individuals.

**Experimental set-up.** Newly born offspring were randomly assigned to one of three treatments directly after birth (Fig. 1).

In the 0-day treatment, directly after birth, four individuals from a single brood were transferred into a single 30 cm × 30 cm × 30 cm square tank, which was covered on all sides with black foil to limit outside disturbances. The tank was divided into four equally sized quadrants using a blue filter sponge. Each compartment housed a single individual and the sponge divider allowed water exchange between the four compartments but no visual or direct interactions of the fish. Such a design allows olfactory communication among physically isolated fish which we know is important in this species[37,38] and should circumvent the development of behavioural abnormalities reported in other studies using teleost fishes that were entirely socially deprived[48,49]. An air-driven filter was integrated into the sponge to maintain water quality. We exchanged 50% of the water on a weekly routine. Water temperature was maintained constantly at 24 °C through room temperature, and as all tanks were placed on a large table at the same height no temperature variation greater than 1 °C was observed in our weekly measurements. The tank contained no gravel or any other substrate. To (i) avoid size (growth) differences between individuals due to competition for food (we observed no differences in body size between individuals, see results) and (ii) maintain good health of all fish throughout the experiment (only 2 out of 96 experimental individuals died before phenotyping), fish were fed *ad libitum* with live *Artemia*-nauplii three times a day as well as with commercially available dusted flake food (TetraMin) twice a day. Such a feeding regime follows standard protocols for common garden experiments in mollies[50,51]. After 7 weeks of isolation, each fish was individually phenotyped for its exploration and activity (see ref. 52 for a similar protocol). We chose exploration and activity patterns as our target trait since locomotion is of central importance for all non-sessile animals and individual differences in this trait are thought to have substantial ecological consequences[53]. Furthermore, activity is influenced by the social environment in the Atlantic molly (*Poecilia mexicana*), one of the parental species of the Amazon molly[32].

In the 7-day treatment, four individuals from a single brood were transferred to a similar square tank as described for the 0-day treatment. However, fish were reared as a group without a sponge divider for 7 days, allowing unlimited direct social interactions (for example, social hierarchies[31]). Afterwards, fish were isolated via a sponge divider and reared separately for another 6 weeks before being phenotyped. Maintenance was as described for 0-day treatment.

Finally, in the 28-day treatment, four individuals from a single brood were transferred to a similar square tank as described for the 0-day treatment. Fish were then reared as a group without a sponge divider for 28 days, allowing unlimited direct social interactions. Afterwards, fish were isolated via a sponge divider and reared separately for another 3 weeks before being phenotyped. Maintenance was as described for 0-day treatment.

We repeatedly assayed all individual's behaviour at the age of 7 weeks. To limit the possibility of differences in energy status among individuals affecting our behavioural measures, we did not feed the morning before the tests, which were always done between 10:00 hours and 13:00:00 hours. We used an open-field arena that consisted of a circular tank (48.5 cm in diameter, made of white plastic) filled with system water to a depth of 3 cm. Lighting was provided from neon tubes positioned at the room ceiling, which helped to avoid shadows or reflections within the tank. In each trial, a single fish was introduced into an opaque plastic cylinder in the centre of the arena and let to acclimate for 3 min. Then, the cylinder was

carefully removed and we videotaped fish exploration and activity with a webcam for the next 6 min. After that, we transferred the fish back into its rearing compartment. We exchanged the water in the open-field arena after every trial to exclude any effect of released chemicals on subsequently tested fish. Measurements were repeated every other day until we completed four measurements per individual. Fish were tested in random order, so that no fish was always tested as first or last in a day. Videos were analysed using the automated video tracking software EthoVision XT Version 10.1 (Noldus Information Technologies, Inc.), thus, experimenters were blinded about individual fish treatment identity. Position scoring started 10 s after the fish was released from the cylinder and we measured the total distance swam within the following 5 min. After completing the four activity measures, standard length of all fish were measured to the nearest 0.1 mm. Two experimental individuals died before phenotyping, resulting in final sample sizes of: 0-day treatment $N = 31$; 7-day treatment $N = 31$; 28-day treatment $N = 32$. Choosing our sample size, we followed the recommendations in ref. 54, and the number of animals as well as the number of repeats per individual is large enough to yield sufficient power in detecting repeatability if present[54]. The reported experiments comply with current German law approved by LaGeSo Berlin (GO124/14 to D.B.).

**Statistical analysis.** Our behavioural measure, total distance swam, was square-root transformed before analysis to meet assumptions of normality and homogeneity of variance. We first tested for differences in body size (standard length) among fish from our different treatments using a linear mixed model with standard length as the response variable and treatment as the fixed effect; mother identity was included as a random effect. To investigate whether levels of individual behavioural variation differed across treatments, we used a linear mixed model including the covariates 'observation (trial 1 to 4)' to account for behavioural changes over the course of the experiment[55] and 'standard length' to account for individual body size differences as well as 'treatment' as a fixed factor. We included treatment-specific variance estimates for individual and residual variance. Additionally, we included the global (that is, not treatment specific) random effect of mother identity. The resulting variance components were used to estimate the proportion of variance attributable to the individual, that is, behavioural repeatability[54,56]. We did not include any variation attributed to mother identity in these repeatability estimates as this random effect was not well supported by the model (see Results section) and any variation attributable to mother identity would be split across all treatments (due to our split-brood design). A significant repeatability estimate is interpreted as evidence of consistent individual differences. Significant differences in variance components and repeatability between the treatments can be assumed when the 95% confidence intervals of the estimates do not overlap. We additionally tested whether there was overall support for the treatment-specific variance estimates (that is, whether among-individual variance was different across treatments) by comparing this model to a model where we did not allow the among-individual variance in individual intercepts to vary across treatments (that is, we fit one variance estimate for individuals across all three treatments). We compared the resulting deviance information criterion (DIC) of each model, and if the DIC was reduced by greater than three by including the treatment-specific random effect, we considered this statistical support that the treatment-specific variance estimates better fit our data. Finally, in preliminary analyses, we tested for the possibility that individuals within each treatment exhibited differing plasticity over the repeated testing (that is, random slopes/regression). We did this by running a separate model for each treatment where we included random intercepts and slopes for individuals and mothers. However, there was no evidence that inclusion of the random slopes increased model fit (ΔDIC < 1), and actually impaired model convergence, therefore, we did not retain these terms in our final model (see SI data, Table 1). Additionally, there was no evidence for an interaction between treatment and body size, or treatment and observation (data not shown).

For all analyses, we used Markov chain Monte Carlo estimation, assuming a Gaussian error distribution with the MCMCglmm package in R v3.1.3 (ref. 57). We used parameter-expanded proper priors and preliminary analyses indicated that our results were not sensitive to changes in prior specification. To ensure model convergence, we ran five chains for each model with 500,000 iterations, 1,000 burn-in and thinning every 100 iterations. We visually checked the posterior density plots to ensure proper model mixing and convergence.

**Data availability.** All data are accessible via dryad respository[58].

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

## Acknowledgements

We thank David Lewis and Marcus Ebert for help with animal care as well as all members of the B-Types group. We are especially grateful to Johanna Tiefenbacher and Sophie Schultz who helped with data collection. Furthermore, we thank Manfred Schartl (University of Würzburg) for providing us with individuals of the Amazon molly. Drawings of a female molly in Fig. 1 are courtesy of Madlen Ziege. We received financial support from the Leibniz Competition (SAW-2013-IGB-2) and from the DFG (BI 1828/2-1; LA 3778/1-1).

## Author contributions

All authors designed the study. D.B. performed the experiments, K.L.L. analysed the data. All authors wrote the manuscript.

## Additional information

**Competing interests:** The authors declare no competing financial interests.

