## [Peer Review File · Nature Communications]

Reviewers' Comments:

Reviewer #1 (Remarks to the Author):

This article has much going for it. It describes a straightforward and relatively simple experiment designed to determine whether consistent individual differences in a biologically relevant behavior (activity rate) develop when isogenetic animals are reared under controlled conditions (in isolation), and whether such differences are affected if comparable subjects are reared instead under different social conditions. The authors present findings that suggest that consistent differences in activity rate (measured over a relatively short period of time) develop even for isogenetic animals reared in isolation, and that social rearing conditions have little effect on their results. To my knowledge, this is the first time that empiricists have attempted to do this, although there are of course many studies showing that the trait values expressed at a given age vary among isogenetic animals reared under carefully controlled conditions, as well as a handful of studies that have looked at the consistency of individual differences in behavior over extended (ontogenetic) periods of time in isogenetic animals.

However, there are also a few problems with the current article. First, the authors tend to 'oversell' their results. For instance, they conclude that their findings show that "individuality emerges under conditions aimed to eliminate all factors causing behavioural differentiation" (line 16), but they don't seem to have considered two factors (maternal effects, inherited epigenetic factors) previously shown to affect the development of individual differences in behavioral and other traits in animals (see 1, below). I am also a bit concerned by the methods they used to estimate the temporal consistency of behavior in this study (see 3 and 4). If these and the other concerns noted below can be addressed, this would be a worthwhile contribution to the literature.

1. Although traditionally (e.g. in classical quantitative genetics) individual differences are attributed to either genes or experiences that occur to individuals after birth or hatching, more recently considerable attention has been paid to other factors, including maternal (or parental) effects and inherited epigenetic factors.

In the present experiment, the subjects were derived from 7 different mothers, which may have varied from one another with respect to their ages, experiences before parturition, and various morphological, physiological and behavioral traits. Indeed, the authors' data suggests that

isogenetic animals can develop different behavioral traits even if they are reared under the same, carefully controlled, conditions, which implies that the mothers of the subjects of these experiments may have also differed from one another in various ways.

The authors may have the data required to determine whether there was any indication that the behavior of the experimental subjects varied as a function of the identity of the mother, although their relatively small sample sizes might not be adequate to detect a 'significant' effect of maternal identity on behavior, given the number of other factors being tested. But at the very least, it would be useful to know the effect size for such maternal effects (significant or not).

Unfortunately, it would not be as easy to test for the effects of inherited epigenetic factors on the individual differences reported in this study. However, studies of other species indicate that such factors can contribute to the development of individual differences in isogenetic individuals (e.g. mono-zygotic twins, see Daxinger & Whitelaw, 2012, *Nature Reviews Genetics* doi:10.1038/nrg3188). But the authors should at least acknowledge this rapidly growing literature, which demonstrates that genes, post-natal experiences and maternal effects are not the only mechanisms that contribute to individual differences in behavioral and other traits.

2. The authors should acknowledge other recent studies on the development of personality in isogenic animals. For instance, Edenbrow & Croft have published a series of articles on the development of personality in an isogenetic fish (the mangrove killifish) (e.g. see 2011, *Animal Behaviour* 82: 731-741, 2013, *Oikos* 122, 667-681). They have investigated differences in the development of behavior among isogenetic fish reared under different conditions (including in isolation versus in the presence of conspecifics). In addition, they measured the repeatability of several personality traits, although in their case they measured trait values at extended intervals across ontogeny, as opposed to across a period of a few days, so not surprisingly, their estimates of repeatability were low.

3. I am a bit concerned about the methods used to estimate repeatability in the current article. Over the years, many authors (including Nakagawa & Schielzeth 2010, which was cited in this article) have pointed that if trait values change as a function of time or context, those factors should be included in the indices used to compute R. In addition, if trait values vary differently among different individuals as a function of time or context, then R can't be used as an index of the consistency of those traits across all of the observations (see Biro & Stamps 2015, *Animal Behaviour* 105, 223-230 for a recent review of this topic).

I raise this issue because Figure 2 in the current article suggests 1) that distance swam increased as a function of time for most the subjects under all three rearing conditions, and 2) that the slopes of the relationship between time and distance swam varied among the subjects within each set of rearing conditions. However, I was unable to find any indication in the analyses of R that

the authors included a fixed factor for change in scores as a function of time (here, observation number), or whether they asked whether relationships between scores and time varied across the individuals in each treatment.

3. Related to previous point... if activity values did change as a function of time in at least some of the subjects, then the authors would be studying temporal behavioral plasticity (changes in activity as a function of time), as well as individual differences in activity levels. And since the fish were tested in the same (novel) arena on each of the four occasions, a general tendency for activity to increase from observations 1 to 4 could be interpreted as acclimation to the situation in the arena. But if there WAS a tendency for behavior to change over time, then this particular protocol would not be optimal for a study whose goal was to measure individual differences in activity rates. Instead, it might have been better to measure the subjects' activity after all of them had had an opportunity to become thoroughly familiar with the test arena, or (alternatively), figure out a way to measure their activity in their familiar (home) tank.

4. The authors might consider providing a better rationale for testing the hypothesis that social rearing conditions would affect activity rates in this species. Theory predicts that social conditions early in life would affect the development of activity rates later in life in species in which social conditions vary across time or space, and in which different levels of activity are optimal under different social conditions later in life. Is there any reason to suppose that these conditions might apply to these mollies? Alternatively, are there empirical studies of this or related species that indicate that behavioral traits (including activity rates) vary as function of social rearing conditions?

5. In general, the literature cited seems a bit dated. Of the 25 references cited, only 5 were published since 2010. The empirical and theoretical literature on the development of individual differences in behavior in animals has exploded in the last few years, and this article would be improved if the authors could consult the most-recent work on the topic (e.g. see items 1-3, above).

Reviewer #2 (Remarks to the Author):

Review of: Bierbach et al.-On the inevitability of behavioural individuality

Review by: Dan Blumstein

This is a really exciting paper that, if the results are robust and valid, provides an exciting and new contribution to the personality literature. By raising genetically- homogenous fish in a set of tightly controlled conditions, the authors assert that they were able to test two fundamental assumptions of personality research.

Given their strong assertions that their design is bullet proof, I have picky questions about the design. I feel certain that they can address them-either by clarifying my confusion or by conducting additional analyses. If adequately addressed, I believe these are highly important findings that help us better understand, as the authors put it, the inevitability of individuality. I've got a concern with the 'non-social' housing. Olfactory signals were potentially shared between animals. Wouldn't it have been better to house animals individually with only their OWN olfactory signals to properly isolate the effect of genes and environment? Doesn't this provide some variation in experience across and within treatments? For instance, one individual could excrete glucocorticoids and couldn't this influence others' responses? I only suggest this because the authors assert that this design eliminated both genetic and experiential effects but I'm not sure that there was not variation in experience. Yes, I realize that all individuals in that block MIGHT have shared the same olfactory experience, but I guess I'd like to see this tested for specifically. Or, do the authors think that by housing them with olfactory exposure only, it controls for 'something' when making the comparison to social housing? Again, I'd like the blocks to be tested.

Fig. 2: Shouldn't these intercepts be greater than or equal to 0? I'm a bit unclear about what exactly was done with the analysis that led to this graph. The say smoothed, but does this analysis estimate BLUPS. Shouldn't it be constrained to positive values only?

L142: I don't quite understand why the split brood design controlled for maternal effects? Assuming there was variation in mothers' investment strategies, or a mother's prior experiences were transferred to her young (epigenetically or some other way), how does splitting the brood eliminate maternal effects, particularly without testing for brood or group effects? I need a little more explanation. Perhaps more importantly, why not decompose variance components to estimate the maternal effects? If they were not different than 0 then wouldn't this confirm your assertion that you controlled for them?

Essentially, most of the above concerns refer to their first conclusion. I find their second conclusion fascinating! While I think the design is adequate to address that conclusion, I will say that their dependent variable (open field exploration) might not be the best variable to test for social traits and it would have been nice if they tested for the effects of social behavioral types with a social assay. Might be worth nothing that. It's also worth noting that the magnitude of the personality effect is on the order of that reported in many other studies of non-genetically homogenous animals raised under highly controlled conditions.

Regardless, these results are extremely interesting and well worthy of publication after some clarification/justification/additional analyses.

Reviewer #3 (Remarks to the Author):

General comments:

This behavioural experiment addresses a question of general importance in the growing field of animal personality. What are the causes of the development of behaviour individuality? In general I found this manuscript a very interesting read. Moreover, the idea to use a clonal species to remove genetic effects is very interesting, and in many respects the experimental design is well thought out. That said, I still have concerns over a number of aspects of this study, including the validity of the basic predictions, the methodology, and the interpretation of the results.

The authors first prediction is that individuals reared under identical environmental conditions will develop identical behavioural phenotypes. While this prediction may hold theoretically, it is in practical impossible to create identical environmental conditions even under controlled laboratory conditions. Moreover, even if the experimental design of this study is credible in many respects, the authors did not use the best possible design to create identical conditions. First, in the 0-day treatment water exchange was allowed between the four compartments. This means that olfactory communication was possible between individuals, thus environmental influences were not minimized. This may have been less of a problem if the Amazon molly did not use olfactory communication. However, a recent study by Reding and Cummings (2016 (not cited by the authors)) shows that olfactory communication is used by Amazon mollies for recognizing conspecifics and may also be used by females to avoid heterospecific males. Thus, chemical communication is likely an important aspect of the social environment of this species. Second, the fish in all treatments were fed two types of food: commercial Tetramin flakes, and live *Artemia*-naupli. Using two food types provides individual fish with a potential for developing food preferences for either live or flake food over ontogeny. Also, the distribution and movements of a live food species cannot be standardized - thus the variation of the feeding environment is increased - and all this may affect the resulting activity patterns. While these comments may seem picky (which they are) they are nevertheless relevant in a paper when environmental control is highlighted as critical for the interpretation of the results.

The authors second prediction is mainly based on recent general reviews suggesting that increasing social experience during ontogeny should promote individuality through a variety of social processes (for example frequency-dependence, niche specialization and reputation formation). While this may be valid for some species, I would claim that the individuality-promoting effects of social experience should be highly dependent on the specific ecology and sociality of the species studied. The authors do not provide any information on the social behaviour of their study species, which they should do because it is critical for generating predictions based on sociality. When I checked this out it turns out that Amazon Molly is a highly social species that lives in large all-female shoals (Schlupp 2009 authors own ref.). Even if female-female aggression occurs in these shoals they are also likely to function as centers of social information. Recent studies suggest that use of social information in such groups can lead to conformity in behaviour (reduced individuality) because individuals will adopt the behaviours

of others when private information is lacking or less reliable than public information (e.g. Pike & Laland 2010). Thus, I would be very hesitant to predict that social experience will increase individuality in Amazon mollies.

Specific comments:

Main text

L. 33-36 and 62-64. I am skeptical to prediction 2 (see above).

L. 39-40. Basic information on the ecology and social behaviour of the study species is missing and should be added since such information is critical for generating predictions and interpreting results.

L.69-70. I do not think that the design here is optimal to minimize environmental factors (see above).

L. 76-78. This tendency is consistent with the alternative/complementary hypothesis that social experience leads to conformity reducing individuality (see above).

Table I. Table text needs to specify what the values within parentheses represent. If they are confidence limits there appears to be something wrong with the values in the upper-left corner? (33.1 (12.4, 17.1)). Same thing in Extended Data table 1.

L. 97-98 and 109-111. I would question that prediction 2 is fundamental (see above).

L. 105-106: The authors should recognize and cite previously published related animal personality studies that have been conducted on other clonal fish species, for example by Edenbrow et al. (2013), especially since their results are consistent with yours: i.e. social experience did not increase repeatability of behaviour.

L. 114-117. I agree with this point and think it is an important one.

Methods

L. 138-139. Captive breeding/domestication in general is known to have considerable evolutionary effects on the phenotype, where behavioural traits can change rapidly, often driven by directional selection. This is another critical aspect to evaluate the ecological relevance of this study. In the special case of genetically identical clonal fish, however, it is unclear to me if at all/how much this population is changed from the wild founder population? If you have any

supporting information on this it would be valuable.

L.151-153. Water exchange allows for chemical communication (see comment above).

L.151-152. Did you check that light intensity and water flow was the same in all four quadrants?

L. 155. Typo: exchanged 50% of the water?

L.157. Inclusion of two food types, including "unpredictable" live food, increases environmental variation (see comments above).

L.171-172 and 178-181. Please inform whether you starved the fish for a standardized period before testing for activity during the four trials. If not, behavioural variation among individuals can be generated simply by random variation in energy status affecting motivation and activity levels at the onset of the trials.

L. 172-173. Note that seemingly small differences in test arena design can affect the outcome of activity trials, including whether treatments, populations etc. differ or not. This may resemble what you call micro-environmental variation (L. 116) - but it is actually measurable, at least to some extent (Näslund et al. 2015).

Extended data figure 1: The presentation of statistics in Fig 1. and in the fig. text does not seem to match and this is not explained in the figure text. It appears that you use means and CI in the fig text, whereas the fig is a box-plot with non-parametric median and inter-quartile range etc. The box plots suggest that the data is skewed which would indicate that the requirements for parametric variance analyses are not met, or that the data needs transformation. Please provide information to support the use of parametric analysis, or use a non-parametric alternative. Revise Fig. 1 accordingly so either a parametric or non-parametric representation of the data is used.

My references:

Edenbrow & Croft 2013 *Oikos* 122: 667-681

Näslund et al. 2015 *Ethology* 121: 556-565.

Pike & Laland 2010. *Biol Lett* 6: 466-468

Reding & Cummings 2016. *Behav Ecol* 27: 411-418

Reviewer #4 (Remarks to the Author):

I enjoyed the premise of the study, and thought it was well written (many I review are not!). Background, logic, predictions and methods were all well laid out despite space restrictions. The question being tested is an interesting one, and contrasts with many studies I've seen on the topic of animal personality, as being topical and novel.

I am generally supportive, but at same time raise some concerns that I think make this study not quite as clear or convincing as it might have been. My two main issues to raise are (a) whether or not the study was really successful in creating environments that could not have generated individual differences prior to the social manipulation and (b) sufficiency of data/analyses for a powerful test of the predictions, under the assumption that my concern in (a) is not a problem. If both issues are in fact a problem, or likely, then I think this might lead one to question the study more generally.

Concern (a) - initial conditions

Although the study "was deliberately designed to eliminate all factors causing behavioural individuality", I am not fully convinced this was the case. Certainly, many factors were eliminated, but others remain.

First, for example, water temperature was apparently maintained at 24C via room temperature, but no variation information was given, and should be included. However, even in "constant" temperature rooms, there is always systematic variation in temperature around the room AND if fish are held on shelves, then temperatures are often 2 to 3C different from bottom to upper shelves. I have worked in several such rooms and have always found substantial spatial/temporal variation in temperature. The relevance of temperature is that it directly affects metabolism, behaviour and growth (Biro et al. 2010, Salinas and Munch 2012) and has the potential to prime individuals and set them on different behavioral/LH trajectories after 7 weeks in the lab.

Second, my reading of the ms indicates that fish were held in groups of four that shared the same water, and thus water quality. This means that water quality differences among groups (tanks) could make individuals within a tank experience different conditions than those in other tanks, just due to variation in ammonia or small differences in food provided. This possibility could be tested for in the model (see below on stats) as a random effect (subject = group), and then the other random effect would be individual nested in group (subject = ID(group)).

Concern (b) - Stats and interpretation

The data analysis indicates proficiency in statistics, but there is still one minor and one 'major' problem I see with it.

First, and most importantly, there is clear evidence that individuals differ in responses over time (see Fig 2) but their model assumes that reaction norms are parallel and increasing together at the same rate of apparent habituation (there is a mean level effect of observation). This violates assumptions of the model and renders inference questionable.

That is, the key comparisons the authors trying to make are those comparing R values between the three treatment groups. However, these R values are technically not valid and the interpretation of them is that individual differences are maintained. In other words, the evidence (data in fig 2) seems to indicate that 'personality' changes over the four assays (rank orders change) or rather perhaps that there is no evidence for personality. These issues on the use and interpretation of R are perhaps old, but were raised recently by Biro and Stamps 2015.

The authors conclude however that "First, we find that substantial behavioural individuality emerges even in the absence of any initial differences between individuals housed under identical conditions." If repeated measures over four days seems to indicate substantial rank-order changes in activity (Fig. 2), then how can one come to this conclusion? The authors should also keep in mind the assumptions of the random effects, which assume normality of blups (predicted means) which may not be valid here.

When I look at fig 2 I think immediately that the analysis should contain a random slope effect with respect to observation number, in addition to the random intercept effect of ID. But, with only 4 measures per individual, and 30 fish, it will be difficult indeed to test for a random slope effect within each treatment; if there is no support for a random slope effect(s), then the data should be more carefully interpreted in light of my comments above. Even in the absence of random slope effects (e.g. if fig 2 showed parallel reaction norms), this study has relatively low power and precision and so comparing variance parameters among treatments, and finding no differences is not particularly strong evidence that social context has no effect.

On a less critical note, I think the overall data set can and should be analysed together. Random effects can be fit specific to each treatment given that individuals are uniquely nested within each treatment. That is, and separate random intercept and residual variance can be fit for each treatment.

Was data transformed? Should be!

Minor points

In the introduction I did not see reference to Edenbrow (Edenbrow and Croft 2013) which seems relevant to cite in the introduction.

Line 176: trials consisted of a 3min acclimation, then activity measurement for 6min. Why were these values chosen? Pilots? Or arbitrary?

173: what "system water"?

Biro, P. A., C. Beckmann, and J. A. Stamps. 2010. Small within-day increases in temperature affects boldness and alters personality in coral reef fish. *Proc. R. Soc. Lond. B* 277:71-77.

Edenbrow, M. and D. P. Croft. 2013. Environmental and genetic effects shape the development of personality traits in the mangrove killifish *Kryptolebias marmoratus*. *Oikos* 122:667-681.

Salinas, S. and S. B. Munch. 2012. Thermal legacies: transgenerational effects of temperature on growth in a vertebrate. *Ecology Letters* 15:159-163.

Response to Reviewers:

Reviewers' comments:

Reviewer #1 (Remarks to the Author):

This article has much going for it. It describes a straightforward and relatively simple experiment designed to determine whether consistent individual differences in a biologically relevant behavior (activity rate) develop when isogenetic animals are reared under controlled conditions (in isolation), and whether such differences are affected if comparable subjects are reared instead under different social conditions. The authors present findings that suggest that consistent differences in activity rate (measured over a relatively short period of time) develop even for isogenetic animals reared in isolation, and that social rearing conditions have little effect on their results. To my knowledge, this is the first time that empiricists have attempted to do this, although there are of course many studies showing that the trait values expressed at a given age vary among isogenetic animals reared under carefully controlled conditions, as well as a handful of studies that have looked at the consistency of individual differences in behavior over extended (ontogenetic) periods of time in isogenetic animals.

Thank you very much for the positive evaluation of our manuscript!

However, there are also a few problems with the current article. First, the authors tend to 'oversell' their results. For instance, they conclude that their findings show that "individuality emerges under conditions aimed to eliminate all factors causing behavioural differentiation" (line 16), but they don't seem to have considered two factors (maternal effects, inherited epigenetic factors) previously shown to affect the development of individual differences in behavioral and other traits in animals (see 1, below). I am also a bit concerned by the methods they used to estimate the temporal consistency of behavior in this study (see 3 and 4). If these and the other concerns noted below can be addressed, this would be a worthwhile contribution to the literature.

1. Although traditionally (e.g. in classical quantitative genetics) individual differences are attributed to either genes or experiences that occur to individuals after birth or hatching, more recently considerable attention has been paid to other factors, including maternal (or parental) effects and inherited epigenetic factors.

In the present experiment, the subjects were derived from 7 different mothers, which may have varied from one another with respect to their ages, experiences before parturition, and various morphological, physiological and behavioral traits. Indeed, the authors' data suggests that isogenetic animals can develop different behavioral traits even if they are reared under the same, carefully controlled, conditions, which implies that the mothers of the subjects of these experiments may have also differed from one another in various ways.

The reviewer raises a very relevant point and we agree that the use of multiple mothers indeed introduces the possibility of differences in maternal effects across experimental individuals. We tried to control and test for maternal effects (and epigenetic effects) in a number of ways. First, the mothers that we used were themselves sisters from the same brood. Thus we know that they are all the same age, from the same mother, and had experienced similar environmental conditions during their lives. We then additionally split the broods of these mothers across our three different treatments, which while not removing the possibility for maternal effects, at least ensure that any effects are not biased within just a single treatment. Finally, (as a response to the comment below) we did also include 'Mother ID' as an additional random effect in our models. We found that this term generally was not well supported by the information criterion and in fact made our model perform more poorly in terms of convergence, so we initially removed it. However, the reviewer is right that mother identity is a pertinent source of variation and so we have now explicitly included 'Mother ID' as a random effect in our models. Our results and their interpretation are not influenced by this additional factor (see the new Table 1, SI table 1). Therefore, we feel confident that we were able to successfully experimentally limit, and then statistically control for any variation introduced by the mother's identity.

The authors may have the data required to determine whether there was any indication that the behavior of the experimental subjects varied as a function of the identity of the mother, although their relatively small sample sizes might not be adequate to detect a 'significant' effect of maternal identity on behavior, given the number of other factors being tested. But at the very least, it would be useful to know the effect size for such maternal effects (significant or not).

In light of this, we performed additional statistical analysis, including 'Mother ID' as a random factor (lines 257ff):

“To address our main research question of whether levels of individual behavioural variation differ across treatments, we used a linear mixed model including the covariates ‘observation (trial 1 to 4)’ to account for behavioural changes over the course of the experiment⁵⁴ and ‘standard length’ to account for individual body size differences as well as treatment as a fixed factor. We included treatment-specific variance estimates for individual and residual variance. Additionally, we included the global (i.e., not treatment-specific) random effect of mother identity. The resulting variance components were used to estimate the proportion of variance attributable to the individual, e.g., behavioural repeatability^{55,56}. We did not include any variation

attributed to mother identity in these repeatability estimates as this random effect was not well supported by the model (see results) and any variation attributable to mother identity would be split across all treatments (due to our split brood design)."

Unfortunately, it would not be as easy to test for the effects of inherited epigenetic factors on the individual differences reported in this study. However, studies of other species indicate that such factors can contribute to the development of individual differences in isogenetic individuals (e.g. mono-zygotic twins, see Daxinger & Whitelaw, 2012, Nature Reviews Genetics doi:10.1038/nrg3188). But the authors should at least acknowledge this rapidly growing literature, which demonstrates that genes, post-natal experiences and maternal effects are not the only mechanisms that contribute to individual differences in behavioral and other traits.

We now explicitly mention the possibility of epigenetic effects at lines 182ff:

"To further limit the potential for differences among mothers, e.g., in maternal effects, these gravid females were all sisters from the same brood ensuring all individuals were the same age, from the same mother, and had experienced similar conditions for their entire lives."

and 129ff:

"Most current explanations of behavioural individuality are based on either of three factors (or an interaction of those)^{8-17,34,35}: (i) inherent (i.e. genetic and epi-genetic) differences between individuals, (ii) between-individual differences in 'major' ecological factors like mortality risk, predation regime, resource abundances and competitive regime and/or (iii) direct social interactions."

In addition, we now try to explain our results in more detail, mentioning also micro-environmental differences, stochastic processes and developmental instabilities as possible causes for the observed individual differences (see lines 137ff):

"First, as we have stressed above, it is practically impossible to provide identical environmental conditions to different individuals and it is likely that different individuals in our experiment experienced different micro-environments, for example slight differences in water temperature, olfactory signals or distribution of and preferences for certain prey items. The observed behavioural individuality, in turn, might be understood as a (potentially adaptive) response to such differences. Second, the observed behavioural individuality might not be caused by micro-environmental differences but rather be an inevitable outcome of the development of the behavioural phenotype, that is we may have captured the natural stochasticity that occurs during the developmental process³⁶⁻⁴⁰ (e.g., developmental instability, stochasticity in gene expression patterns, bet-hedging). Both explanations provide a substantial challenge to the current research paradigm that aims to explain behavioural individuality with differences in genetic make-up, major ecological factors and/or direct social interactions."

2. The authors should acknowledge other recent studies on the development of personality in isogenic animals. For instance, Edenbrow & Croft have published a series of articles on the development of personality in an isogenetic fish (the mangrove killifish) (e.g. see 2011, Animal Behaviour 82: 731-741, 2013, Oikos 122, 667-681). They have investigated

differences in the development of behavior among isogenetic fish reared under different conditions (including in isolation versus in the presence of conspecifics). In addition, they measured the repeatability of several personality traits, although in their case they measured trait values at extended intervals across ontogeny, as opposed to across a period of a few days, so not surprisingly, their estimates of repeatability were low.

We apologize for not having included these relevant studies. The Edenbrow and Croft papers are highly relevant and our study builds nicely on them. Importantly, the key focus of the Edenbrow and Croft papers is on linking behavioral individuality to genetic and environmental differences. In contrast, our study explicitly controls for both genetic differences among individuals and differences in key socio-ecological factors. We have now made efforts to integrate these papers into our manuscript which we think places our results in a broader context (lines 121ff):

“Previous work on humans and clonal animals has already demonstrated that differences in genotypes^{29,30} and/or social environments³¹⁻³³ can by themselves generate individuality. Our study thereby builds on this previous work to explicitly control for both genotypic differences and variation in the predicted socio-ecological factors to test whether individuality still emerges.”

3. I am a bit concerned about the methods used to estimate repeatability in the current article. Over the years, many authors (including Nakagawa & Schielzeth 2010, which was cited in this article) have pointed that if trait values change as a function of time or context, those factors should be included in the indices used to compute R. In addition, if trait values vary differently among different individuals as a function of time or context, then R can't be used as an index of the consistency of those traits across all of the observations (see Biro & Stamps 2015, *Animal Behaviour* 105, 223-230 for a recent review of this topic).

I raise this issue because Figure 2 in the current article suggests 1) that distance swam increased as a function of time for most the subjects under all three rearing conditions, and 2) that the slopes of the relationship between time and distance swam varied among the subjects within each set of rearing conditions. However, I was unable to find any indication in the analyses of R that the authors included a fixed factor for change in scores as a function of time (here, observation number), or whether they asked whether relationships between scores and time varied across the individuals in each treatment.

The estimates of repeatability were calculated using the variance components from a model that included both individual standard length (as a measure of body size) and observation number, as suggested by the reviewer and the full model was reported in the supplemental. We now show our full model in the main text (table 1) and acknowledge the work by Biro and Stamps (AB 2015) in the description of the statistical analysis (lines 257ff, reference No. 54):

“To address our main research question of whether levels of individual behavioural variation differ across treatments, we used a linear mixed model including the covariates ‘observation (trial 1 to 4)’ to account for behavioural changes over the course of the experiment⁵⁴ and ‘standard length’ to account for individual body size differences as well as treatment as a fixed factor.”

Additionally, the reviewer raises a good point about the necessity to check for individual variation in plasticity (slopes). The lines shown in initial Figure 2 were not the predicted slopes or intercepts from the models, but rather just the raw data that we smoothed for illustrative purposes. We in fact did test for differences in slopes with our data, but found no evidence that the inclusion of random slopes improved our model (and in fact it appeared to make model fit even worse). Therefore we removed these terms from our models. We now report the changes in DIC across the models with these different random structures in the supplemental materials (SI table 1). We additionally have revised Figure 2 to illustrate the predicted estimates for each individual (as opposed to the raw data) to make it more clear that our models predicted the same slopes across all individuals.

3. Related to previous point.... if activity values did change as a function of time in at least some of the subjects, then the authors would be studying temporal behavioral plasticity (changes in activity as a function of time), as well as individual differences in activity levels. And since the fish were tested in the same (novel) arena on each of the four occasions, a general tendency for activity to increase from observations 1 to 4 could be interpreted as acclimation to the situation in the arena. But if there WAS a tendency for behavior to change over time, then this particular protocol would not be optimal for a study whose goal was to measure individual differences in activity rates. Instead, it might have been better to measure the subjects' activity after all of them had had an opportunity to become thoroughly familiar with the test arena, or (alternatively), figure out a way to measure their activity in their familiar (home) tank.

There was indeed a tendency for individuals to increase their activity over the repeated observations, which could be due to some sort of habituation to the testing arena. We attempted to minimize this by giving each individual an “acclimation period” (of 3 minutes, see line 238) after being transferred into the testing arena. However, given that all individuals were exposed to the same arena and appeared to exhibit a similar habituation pattern we do not feel that this overall change in behavior influences our interpretation of our results (i.e., that individuals exhibit significant among-individual variation in behavior even when we control for predicted sources of variation).

4. The authors might consider providing a better rationale for testing the hypothesis that social rearing conditions would affect activity rates in this species. Theory predicts that social conditions early in life would affect the development of activity rates later in life in species in which social conditions vary across time or space, and in which different levels of activity are optimal under different social conditions later in life. Is there any reason to suppose that these conditions might apply to these mollies? Alternatively, are there empirical studies of this or related species that indicate that behavioral traits (including activity rates) vary as function of social rearing conditions?

We thank the reviewer for that valuable suggestion and we have now expanded our rationale for why we would expect activity rates to change in this species with increasing social interactions. In brief, this species lives in large shoal in the wild which raises the opportunity for many social interactions throughout their lives. Furthermore, activity is one of the behavioral traits most important for non-sessile animals and individual differences in activity and exploration behavior are predicted to have substantial ecological and evolutionary consequences. For example, it has been shown that activity patterns vary with social context in the parental species, *Poecilia mexicana* (Bierbach et al 2015 Behav. Ecol., reference No. 28). In this species, adult males that were kept in full isolation from any other fish showed higher activity rates than males

kept within a small group of other males. Also, early life experiences are known to affect adult behavior and dominance formation in our study species (see Laskowski et al. 2016 ProcB, reference No. 27). Thus, there are good reasons to believe that activity rates would be influenced by social interactions also in the Amazon molly (see lines 63ff):

“Amazon mollies occur in large shoals in the wild²⁶ suggesting these animals are regularly exposed to varying levels of social interactions. Previous work has also shown that early social interactions among these individuals can generate long lasting consequences on adult behaviour²⁷.”

and lines 95ff:

*“It is note-worthy that the absolute levels of behavioural variation exhibited by these clonal individuals closely resemble that which we have seen in non-clonal fish measured in a similar way²⁸ (in the Atlantic molly, *Poecilia mexicana*, one of the parental species of *P. formosa*).”*

In addition, part of our rationale for not testing the individuals within a social setting (like for aggressiveness or sociability) is such tests of social behavior might easily affect individual behavior over the repeated testing and thus confound our experimental results (especially for the 0-day fish which had never interacted with another fish before). Please see lines 216ff:

*„We choose exploration and activity patterns as target trait since locomotion is of central importance for all non-sessile animals and individual differences in this trait are thought to have substantial ecological consequences⁵². Furthermore, activity is influenced by the social environment in the Atlantic molly (*Poecilia mexicana*), one of the parental species of the Amazon molly²⁸.”*

5. In general, the literature cited seems a bit dated. Of the 25 references cited, only 5 were published since 2010. The empirical and theoretical literature on the development of individual differences in behavior in animals has exploded in the last few years, and this article would be improved if the authors could consult the most-recent work on the topic (e.g. see items 1-3, above).

Thank you for that suggestion! We expanded our literature cited substantially (there are now 43 references cited in the main text) and now include many of the papers mentioned by the reviewers as well as other works. By this, we hope to sufficiently cover the most recent advances in the topic.

Reviewer #2 (Remarks to the Author):

Review of: Bierbach et al.-On the inevitability of behavioural individuality

Review by: Dan Blumstein

This is a really exciting paper that, if the results are robust and valid, provides an exciting and new contribution to the personality literature. By raising genetically- homogenous fish in a set of tightly controlled conditions, the authors assert that they were able to test two fundamental assumptions of personality research.

Given their strong assertions that their design is bullet proof, I have picky questions about the design. I feel certain that they can address them-either by clarifying my confusion or by

conducting additional analyses. If adequately addressed, I believe these are highly important findings that help us better understand, as the authors put it, the inevitability of individuality.

I've got a concern with the 'non-social' housing. Olfactory signals were potentially shared between animals. Wouldn't it have been better to house animals individually with only their OWN olfactory signals to properly isolate the effect of genes and environment? Doesn't this provide some variation in experience across and within treatments? For instance, one individual could excrete glucocorticoids and couldn't this influence others' responses? I only suggest this because the authors assert that this design eliminated both genetic and experiential effects but I'm not sure that there was not variation in experience. Yes, I realize that all individuals in that block MIGHT have shared the same olfactory experience, but I guess I'd like to see this tested for specifically. Or, do the authors think that by housing them with olfactory exposure only, it controls for 'something' when making the comparison to social housing? Again, I'd like the blocks to be tested.

The reviewer raises an important point. Quite generally, it is practically impossible to provide identical environmental conditions to different individuals even in a highly controlled common-garden design as ours. We believe, however, that this does not hamper the importance of our findings. The key goal of our experimental design was remove all 'major' socio-ecological factors that were previously identified to cause individual variation among animals of the same population (i.e. direct social interactions, differences in mortality risk, predation regime, resource abundances and/or competitive regime). The key finding of our work is that substantial behavioral individuality emerges even in the absence of these factors. We fully agree with the referee that it is likely that different individuals in our experiment experienced different micro-environments, for example, slight differences in olfactory signals, and that such differences may explain the emergence of behavioral individuality in our experiments (we now discuss this issue in detail in our revised manuscript). In our view, however, this does not weaken the importance of our findings. In particular, our findings challenge a long-standing research paradigm within research on behavioral individuality that links the emergence of individuality to (epi-)genetic differences, direct social interactions and/or differences in 'major' ecological factors. In sharp contrast to this paradigm, our findings suggest that individuality is a more general phenomenon, being an inevitable and inherently unpredictable outcome of the development of complex phenotypes.

More specifically, with respect to olfactory cues, we chose to allow the individuals to have access to olfactory cues to ensure a sufficiently high level of "normalcy" in their development. These animals do not occur in isolation in the wild and the removal of all cues might easily result in abnormal behavior (as shown for several fish species when reared in total isolation, see for example references 49 and 50; lines 201ff):

"Each compartment housed a single individual and the sponge divider allowed water exchange between the four compartments but no visual or direct interactions of the fish. Such a design allows olfactory communication among physically isolated fish which we believe to circumvent the development of behavioural abnormalities reported in other studies using teleost fishes that were entirely socially deprived^{49,50}."

Additionally, our goal was that the 0-day treatment acts as a proper control for our social treatments by explicitly controlling for social interactions *per se*. That is, fish in the 0-day treatment were exposed to similar cues as their social treatment siblings

(environment, olfactory cues) but were lacking the direct social interactions. This is important, as the theory on the development of personality differences assumes that direct social interactions in the form of cooperation and competition are key drivers of behavioral individuality.

Finally, to test for the effect of block as suggested by the reviewer, we included this term in our global model, but this random term was not supported by information criterion and severely impaired model convergence (likely because it is partially confounded with Mother ID) therefore we decided to only retain Mother ID in our models. Overall, the inclusion of this additional random effect (block or mother) does not appear to account for a statistically significant portion of the variance (see the change in DIC scores in Table 1, SI Table 1) therefore we feel confident that any potential variation among the blocks does not alter our interpretation of our results (see also comments to Reviewer 1 on our new statistical analysis).

Fig. 2: Shouldn't these intercepts be greater than or equal to 0? I'm a bit unclear about what exactly was done with the analysis that led to this graph. The say smoothed, but does this analysis estimate BLUPS. Shouldn't it be constrained to positive values only?

Our apologies for being unclear. This figure drew the raw data that had been smoothed by a simple linear regression just for illustrative purposes. Based on this comment and the comments of the other reviewers (see above), we have now revised this Figure 2 to show the predicted intercepts (BLUPs) for each individual assuming a similar effect of observation across all individuals (the same slope). Indeed, we tested for the effects of different random (co)variance structures (e.g. random intercepts, slopes across individuals and mothers) and therefore based on the inherently nested design of the study (individuals nested within mothers) and the support from the model comparison, we only retained terms for random intercepts for individuals and mothers (see SI Table 1 for all models).

L142: I don't quite understand why the split brood design controlled for maternal effects? Assuming there was variation in mothers' investment strategies, or a mother's prior experiences were transferred to her young (epigenetically or some other way), how does splitting the brood eliminate maternal effects, particularly without testing for brood or group effects? I need a little more explanation. Perhaps more importantly, why not decompose variance components to estimate the maternal effects? If they were not different than 0 then wouldn't this confirm your assertion that you controlled for them?

We used a split brood design not to remove maternal effects entirely, but rather to ensure that any maternal effects were at least evenly distributed across all treatments (so as not to bias just one treatment, e.g.). But the reviewer, and others raise a valid point – we can easily partition the variance among mothers as well. We have now included these results in our manuscript. Overall, there is limited support that mother identity contributed a statistically significant proportion of the variance (Table 1, SI table 1; based on the width of the 95% CI, and the lack of a decrease in the DIC between models with and without mother identity). Therefore we feel that while the possibility for maternal effects is there in our fish (which in itself is interesting), our results still stand that individual identity accounts for a significant portion of variation even among fish that were housed in such a way as to limit the experience with all factors that we (currently) think cause personality variation.

Essentially, most of the above concerns refer to their first conclusion. I find their second conclusion fascinating! While I think the design is adequate to address that conclusion, I will say that their dependent variable (open field exploration) might not be the best variable to test for social traits and it would have been nice if they tested for the effects of social behavioral types with a social assay. Might be worth nothing that. It's also worth noting that the magnitude of the personality effect is on the order of that reported in many other studies of non-genetically homogenous animals raised under highly controlled conditions. Regardless, these results are extremely interesting and well worthy of publication after some clarification/justification/additional analyses.

We highly appreciate the enthusiasm the reviewer shows for our work! We have also expanded our explanation of why we chose the behaviors we did to help address the concern the reviewer raises (see lines 216ff and answers to reviewer 1).

Additionally we mention other data from (non-clonal) fish to emphasize the reviewer's final point – that the absolute level of variation shown by these genetically identical fish is roughly similar to what would be found even among non-identical individuals (lines 95ff):

*“It is note-worthy that the absolute levels of behavioural variation exhibited by these clonal individuals closely resemble that which we have seen in non-clonal fish measured in a similar way²⁸ (in the Atlantic molly, *Poecilia mexicana*, one of the parental species of *P. formosa*).”*

Reviewer #3 (Remarks to the Author):

General comments:

This behavioural experiment addresses a question of general importance in the growing field of animal personality. What are the causes of the development of behaviour individuality? In general I found this manuscript a very interesting read. Moreover, the idea to use a clonal species to remove genetic effects is very interesting, and in many respects the experimental design is well thought out. That said, I still have concerns over a number of aspects of this study, including the validity of the basic predictions, the methodology, and the interpretation of the results.

The authors first prediction is that individuals reared under identical environmental conditions will develop identical behavioural phenotypes. While this prediction may hold theoretically, it is in practical impossible to create identical environmental conditions even under controlled laboratory conditions. Moreover, even if the experimental design of this study is credible in many respects, the authors did not use the best possible design to create identical conditions.

First, in the 0-day treatment water exchange was allowed between the four compartments. This means that olfactory communication was possible between individuals, thus environmental influences were not minimized. This may have been less of a problem if the Amazon molly did not use olfactory communication. However, a recent study by Reding and Cummings (2016 (not cited by the authors)) shows that olfactory communication is used by Amazon mollies for recognizing conspecifics and may also be used by females to avoid

heterospecific males. Thus, chemical communication is likely an important aspect of the social environment of this species.

Second, the fish in all treatments were fed two types of food: commercial Tetramin flakes, and live *Artemia*-naupli. Using two food types provides individual fish with a potential for developing food preferences for either live or flake food over ontogeny. Also, the distribution and movements of a live food species cannot be standardized - thus the variation of the feeding environment is increased - and all this may affect the resulting activity patterns. While these comments may seem picky (which they are) they are nevertheless relevant in a paper when environmental control is highlighted as critical for the interpretation of the results.

The reviewer raises an important point: quite generally, it is practically impossible to provide identical environmental conditions to different individuals even in a highly controlled common-garden design as ours. We believe, however, that this does not hamper the importance of our findings. The key goal of our experimental design was remove all ‘major’ socio-ecological factors that were previously identified to cause individual variation among animals of the same population (i.e., direct social interactions, differences in mortality risk, predation regime, resource abundances and/or competitive regime). The key finding of our work is that substantial behavioral individuality emerges even in the absence of these factors. We fully agree with the referee that it is likely that different individuals in our experiment experienced different micro-environments (e.g. slight differences in olfactory signals, prey distribution or food preferences) and that such differences may explain the emergence of behavioral individuality in our experiments (we now discuss this issue in detail in our revised manuscript). In our view, however, this does not weaken the importance of our findings. In particular, our findings challenge a long-standing research paradigm within research on behavioral individuality that links the emergence of individuality to (epi-)genetic differences, direct social interactions and/or differences in ‘major’ ecological factors. In sharp contrast to this paradigm, our findings suggest that individuality is a more general phenomenon, being an inevitable and inherently unpredictable outcome of the development of complex phenotypes.

More specifically, with respect to olfactory cues, we chose to allow the individuals to have access to olfactory cues to ensure a sufficiently high level of “normalcy” in their development. These animals do not occur in isolation in the wild and the removal of all cues might easily result in abnormal behavior (as shown for several fish species when reared in total isolation, see for example references 49 and 50; lines 201ff):

“Each compartment housed a single individual and the sponge divider allowed water exchange between the four compartments but no visual or direct interactions of the fish. Such a design allows olfactory communication among physically isolated fish which we believe to circumvent the development of behavioural abnormalities reported in other studies using teleost fishes that were entirely socially deprived^{49,50}.”

Additionally, our goal was that the 0-day treatment acts as a proper control for our social treatments by explicitly controlling for social interactions *per se*. That is, fish in the 0-day treatment were exposed to similar cues as their social treatment siblings (environment, olfactory cues) but were lacking the direct social interactions. This is important, as the theory on the development of personality differences assumes that direct social interactions in the form of cooperation and competition are key drivers of behavioral individuality.

We were also able to test for this possibility by running models that included the random effects of Mother ID and Block. However, these two terms are partially confounded and the inclusion of Block was especially not supported and in fact made model fit worse, therefore we decided to only retain Mother ID in the final model we report in the manuscript. Overall, the inclusion of this additional random effect does not seem to account for a statistically significant portion of the variance (see the change in DIC scores in Table 1, SI Table 1) therefore we feel confident that any potential variation among the blocks does not alter our interpretation of our results.

Finally, with regards to two types of food used, this was done because we wanted to provide all fish *ad libitum* with the best possible diet. The goal of our design was to (a) reduce size (growth) differences due to competition for food in groups (in fact, our treatment fish did not differ in size) and (b) help to maintain good health of all fish throughout the experiment on the long run (which is often difficult with a single food source). So by providing the two types of food sources, we aimed to reduce effects due to differential growth or food competition. Please see lines 210ff:

“Fish were fed ad libitum with live Artemia-nauplii three times a day as well as with commercially available dusted flake food (TetraMin™) twice a day. Although food diversity might be a source for individual behavioural divergence due to differences in food preferences⁵¹, we aimed to provide all individuals with the best available food to ensure that there was no food deprivation throughout our tests.”

The authors second prediction is mainly based on recent general reviews suggesting that increasing social experience during ontogeny should promote individuality through a variety of social processes (for example frequency-dependence, niche specialization and reputation formation). While this may be valid for some species, I would claim that the individuality-promoting effects of social experience should be highly dependent on the specific ecology and sociality of the species studied. The authors do not provide any information on the social behaviour of their study species, which they should do because it is critical for generating predictions based on sociality. When I checked this out it turns out that Amazon Molly is a highly social species that lives in large all-female shoals (Schlupp 2009 authors own ref.). Even if female-female aggression occurs in these shoals they are also likely to function as centers of social information. Recent studies suggest that use of social information in such groups can lead to conformity in behaviour (reduced individuality) because individuals will adopt the behaviours of others when private information is lacking or less reliable than public information (e.g. Pike & Laland 2010). Thus, I would be very hesitant to predict that social experience will increase individuality in Amazon mollies.

We agree with the referee that this is an interesting alternative prediction about the impact of social interactions behavioral individuality in these fish. Indeed, the data we collected is appropriate to directly test this hypothesis. We predicted that increasing scope for social interactions would generate increasing among-individual variation in these fish for a number of reasons. First, a key and robust prediction of a substantial body of theoretical-conceptual work on the emergence of behavioral individuality is that, quite generally, direct social interactions are a key driver underlying behavioral individuality in social species. More specifically, our study species of fish generally form dominance hierarchies in the wild which requires that individuals adjust their behavior to avoid conflict and competition, a mechanism likely to generate individual

differences. Finally, there is some empirical evidence to suggest that increasing time spent within social groups does drive the emergence of behavioral individuality (Laskowski & Pruitt ProcB, ref no. 41). That said, we fully agree with the referee that these two alternative predictions (social experience generates stronger versus weaker among-individual differences) are both appropriate and we now explicitly mention this second alternative hypothesis in lines 159ff.

“Interestingly, the amount of total behavioural variation slightly declines with increasing levels of social experience which might point towards a homogenizing rather than a diversifying effect of living in a stable social environment⁴³.”

Furthermore, we add some information on the social behaviour of these fish at lines 63ff:

“Amazon mollies occur in large shoals in the wild²⁶ suggesting these animals are regularly exposed to varying levels of social interactions. Previous work has also shown that early social interactions among these individuals can generate long lasting consequences on adult behaviour²⁷.”

Specific comments:

Main text

L. 33-36 and 62-64. I am skeptical to prediction 2 (see above).

Please see our justification above.

L. 39-40. Basic information on the ecology and social behaviour of the study species is missing and should be added since such information is critical for generating predictions and interpreting results.

We fully agree. We have now expanded on our introduction of the ecology and natural social behavior of these fish (see above).

L.69-70. I do not think that the design here is optimal to minimize environmental factors (see above).

Please see our response above, and to the other reviewers.

L. 76-78. This tendency is consistent with the alternative/complementary hypothesis that social experience leads to conformity reducing individuality (see above).

Please see our response to the comment above.

Table I. Table text needs to specify what the values within parentheses represent. If they are confidence limits there appears to be something wrong with the values in the upper-left corner? (33.1 (12.4, 17.1)). Same thing in Extended Data table 1.

Our apologies, the values in parentheses were the 95% credibility intervals returned by the MCMC chains. We have now fully described this in the table notes. And the error you noticed was indeed just a type and copying error. This has now been corrected.

L. 97-98 and 109-111. I would question that prediction 2 is fundamental (see above). Please see our response above as well as lines 33ff:

“Second, social processes are predicted to play a key role in generating and maintaining behavioural individuality^{7,11,14-17}. In particular, social processes like frequency-dependence, niche- and role- specialization, and reputation formation are one of the most common and powerful factors used to explain individuality given that they are predicted to promote behavioural individuality even among genetically identical individuals (Prediction 2).”

L. 105-106: The authors should recognize and cite previously published related animal personality studies that have been conducted on other clonal fish species, for example by Edenbrow et al. (2013), especially since their results are consistent with yours: i.e. social experience did not increase repeatability of behaviour.

We fully agree. We now explicitly discuss these studies which help to place our own study in a broader framework. Please see lines 121ff.

“Previous work on humans and clonal animals has already demonstrated that differences in genotypes^{29,30} and/or social environments³¹⁻³³ can by themselves generate individuality. Our study thereby builds on this previous work to explicitly control for both genotypic differences and variation in the predicted socio-ecological factors to test whether individuality still emerges.”

L. 114-117. I agree with this point and think it is an important one.

Thank you! We have now expanded our discussion of alternative mechanisms (such as this one) that might generate the patterns of behavioral variation that we found (lines 136ff):

“We see two potential explanations for the emergence of individuality in our 0-day treatment. First, as we have stressed above, it is practically impossible to provide identical environmental conditions to different individuals and it is likely that different individuals in our experiment experienced different micro-environments, for example slight differences in water temperature, olfactory signals or distribution of and preferences for certain prey items. The observed behavioural individuality, in turn, might be understood as a (potentially adaptive) response to such differences. Second, the observed behavioural individuality might not be caused by micro-environmental differences but rather be an inevitable outcome of the development of the behavioural phenotype, that is we may have captured the natural stochasticity that occurs during the developmental process³⁶⁻⁴⁰ (e.g., developmental instability, stochasticity in gene expression patterns, bet-hedging). Both explanations provide a substantial challenge to the current research paradigm that aims to explain behavioural individuality with

differences in genetic make-up, major ecological factors and/or direct social interactions. “

Methods

L. 138-139. Captive breeding/domestication in general is known to have considerable evolutionary effects on the phenotype, where behavioural traits can change rapidly, often driven by directional selection. This is another critical aspect to evaluate the ecological relevance of this study. In the special case of genetically identical clonal fish, however, it is unclear to me if at all/how much this population is changed from the wild founder population? If you have any supporting information on this it would be valuable.

This is certainly the case with non-clonal animals, however, due to the clonal nature of these mollies we know that at least at a genetic level, these animals are identical to the ones that were initially captured in the wild (personal communication with M. Scharl who collected the animals) and intermittent genetic samplings confirm that all individuals are clones. Please see lines 185ff.

L.151-153. Water exchange allows for chemical communication (see comment above).
Please see our response above.

L.151-152. Did you check that light intensity and water flow was the same in all four quadrants?

We did arrange the outflow of the custom made air-driven filter in a way that each compartment had virtually no direct flow thus preventing between and also within compartment differences in flow regime. Furthermore, light was provided from centrally above the tanks and tanks were covered with black foil to limit outside disturbances (see lines 200ff). Please see also response to reviewer 4.

L. 155. Typo: exchanged 50% of the water?

Apologies, yes we exchanged 50% *“of the water”*. See lines 206ff

“We exchanged 50% of the water on a weekly routine”

L.157. Inclusion of two food types, including "unpredictable" live food, increases environmental variation (see comments above).

We agree that using different food sources gives potential to individual specialization which we now explicitly mention in the manuscript at lines 213ff. That said, we feel the use of different food sources was necessary (please see our detailed response above).

L.171-172 and 178-181. Please inform whether you starved the fish for a standardized period before testing for activity during the four trials. If not, behavioural variation among individuals can be generated simply by random variation in energy status affecting motivation and activity levels at the onset of the trials.

Yes, all animals were starved overnight and trials were performed always between 10 am and 1 pm the next day. Please see lines 233ff.

L. 172-173. Note that seemingly small differences in test arena design can affect the outcome of activity trials, including whether treatments, populations etc. differ or not. This may resemble what you call micro-environmental variation (L. 116) - but it is actually measurable, at least to some extent (Näslund et al. 2015).

- We fully agree, this is a possibility. In our case however, all animals were tested in the exact same test arena.

Extended data figure 1: The presentation of statistics in Fig 1. and in the fig. text does not seem to match and this is not explained in the figure text. It appears that you use means and CI in the fig text, whereas the fig is a box-plot with non-parametric median and inter-quartile range etc. The box plots suggest that the data is skewed which would indicate that the requirements for parametric variance analyses are not met, or that the data needs transformation. Please provide information to support the use of parametric analysis, or use a non-parametric alternative. Revise Fig. 1 accordingly so either a parametric or non-parametric representation of the data is used.

- Apologies, measures of activity were square-root transformed prior to analysis (but we wished to present the raw scores here) to meet the assumptions of normality and homogeneity of errors. Standard length best approximated a Gaussian distribution as confirmed by visual analysis of the model residuals. We accordingly revised Figure 2 in the main text showing now estimates from our model.

My references:

Edenbrow & Croft 2013 *Oikos* 122: 667-681
Näslund et al. 2015 *Ethology* 121: 556-565.
Pike & Laland 2010. *Biol Lett* 6: 466-468
Reding & Cummings 2016. *Behav Ecol* 27: 411-418

Reviewer #4 (Remarks to the Author):

I enjoyed the premise of the study, and thought it was well written (many I review are not!). Background, logic, predictions and methods were all well laid out despite space restrictions. The question being tested is an interesting one, and contrasts with many studies I've seen on the topic of animal personality, as being topical and novel.

Thank you very much!

I am generally supportive, but at same time raise some concerns that I think make this study not quite as clear or convincing as it might have been. My two main issues to raise are (a) whether or not the study was really successful in creating environments that could not have generated individual differences prior to the social manipulation and (b) sufficiency of data/analyses for a powerful test of the predictions, under the assumption that my concern in (a) is not a problem. If both issues are in fact a problem, or likely, then I think this might lead one to question the study more generally.

Concern (a) - initial conditions

Although the study "was deliberately designed to eliminate all factors causing behavioural individuality", I am not fully convinced this was the case. Certainly, many factors were eliminated, but others remain.

First, for example, water temperature was apparently maintained at 24C via room temperature, but no variation information was given, and should be included. However, even in "constant" temperature rooms, there is always systematic variation in temperature around the room AND if fish are held on shelves, then temperatures are often 2 to 3C different from bottom to upper shelves. I have worked in several such rooms and have always found substantial spatial/temporal variation in temperature. The relevance of temperature is that it directly affects metabolism, behaviour and growth (Biro et al. 2010, Salinas and Munch 2012) and has the potential to prime individuals and set them on different behavioral/LH trajectories after 7 weeks in the lab.

Second, my reading of the ms indicates that fish were held in groups of four that shared the same water, and thus water quality. This means that water quality differences among groups (tanks) could make individuals within a tank experience different conditions than those in other tanks, just due to variation in ammonia or small differences in food provided. This possibility could be tested for in the model (see below on stats) as a random effect (subject = group), and then the other random effect would be individual nested in group (subject = ID(group)).

We fully agree, even under the most controlled conditions, there is still bound to be small, stochastic micro-environmental variation. As we have mentioned in other responses, our goal of the study was to control for the ecological factors that current theories predicts will have a predictable impact on individual behavioral variation. Water temperature and quality were measured on a weekly basis which ensured that water temperature stayed within 1° degree during the entire experimental period. We are well aware of potential shelf and tank effects (having seen them in other experiments of ours) and so the tanks used here were explicitly designed to minimize this – all tanks were on the same shelf and lit with careful lighting above to control light intensity (and temperature). Additionally, as other reviewers have pointed out, we have now retained the “mother identity” as a random effect in our models. Mother identity is largely similar to the group effects, given that each mother contributed one brood to the experiment (though one mother did contribute two broods/groups). In all models, there was no support to suggest that mother explained a statistically significant portion of the data (table 1, SI table 1). Therefore we feel confident that any small variations among the different mothers/groups were minimized and do not affect our interpretation of our results.

Concern (b) - Stats and interpretation

The data analysis indicates proficiency in statistics, but there is still one minor and one 'major' problem I see with it.

First, and most importantly, there is clear evidence that individuals differ in responses over time (see Fig 2) but their model assumes that reaction norms are parallel and increasing

together at the same rate of apparent habituation (there is a mean level effect of observation). This violates assumptions of the model and renders inference questionable.

That is, the key comparisons the authors trying to make are those comparing R values between the three treatment groups. However, these R values are technically not valid and the interpretation of them is that individual differences are maintained. In other words, the evidence (data in fig 2) seems to indicate that 'personality' changes over the four assays (rank orders change) or rather perhaps that there is no evidence for personality. These issues on the use and interpretation of R are perhaps old, but were raised recently by Biro and Stamps 2015.

The authors conclude however that "First, we find that substantial behavioural individuality emerges even in the absence of any initial differences between individuals housed under identical conditions." If repeated measures over four days seems to indicate substantial rank-order changes in activity (Fig. 2), then how can one come to this conclusion? The authors should also keep in mind the assumptions of the random effects, which assume normality of blups (predicted means) which may not be valid here.

When I look at fig 2 I think immediately that the analysis should contain a random slope effect with respect to observation number, in addition to the random intercept effect of ID. But, with only 4 measures per individual, and 30 fish, it will be difficult indeed to test for a random slope effect within each treatment; if there is no support for a random slope effect(s), then the data should be more carefully interpreted in light of my comments above. Even in the absence of random slope effects (e.g. if fig 2 showed parallel reaction norms), this study has relatively low power and precision and so comparing variance parameters among treatments, and finding no differences is not particularly strong evidence that social context has no effect.

We have now explicitly included the comparison of models that included random slopes across individuals (and mothers). Overall there was no strong support that individuals significantly differed in their behavior over observation (i.e. random slopes, see SI table 1). Our apologies that Figure 2 was therefore unclear (it initially just showed raw data that had been smoothed for illustrative purposes), we have now revised Fig 2 to show the predicted values for each individual (BLUPs).

We agree that our sample size might not be large enough to excavate weaker differences in random slope effects as mentioned by the reviewer. That said, measuring each of the 90 fish four times is substantially more than most other studies on behavioral individuality (see for example the meta-analysis by Bell et al. 2009, AnimBehav "The repeatability of behaviour: a meta-analysis"). In sum, we are confident that our statistical analysis gives a clear base for our interpretation (see responses above).

On a less critical note, I think the overall data set can and should be analysed together. Random effects can be fit specific to each treatment given that individuals are uniquely nested within each treatment. That is, and separate random intercept and residual variance can be fit for each treatment.

- To the best of our knowledge, it is currently impossible to run one global model with differing parameters for random slopes within each treatment. Therefore, in order to test whether random slopes were supported by the data, we had to run separate models. However, because mother identities are split across all our treatments, we agree with the reviewer, that running a global model is desirable. Therefore we now report in our manuscript the results of this global model (see Table 1, and explanation in the methods Lines 258ff).

Was data transformed? Should be!

- Data has been square-root transformed prior to analysis, which we now explicitly state in Lines 253.

Minor points

In the introduction I did not see reference to Edenbrow (Edenbrow and Croft 2013) which seems relevant to cite in the introduction.

- Our apologies, we have now included this very relevant study (and others on clonal animals). See lines 121ff.

Line 176: trials consisted of a 3min acclimation, then activity measurement for 6min. Why were these values chosen? Pilots? Or arbitrary?

- This was determined by pilot trials which showed that this was an adequate time for the fish to begin swimming normally again after handling.

173: what "system water"?

- Our apologies, this just refers to the fact that we used the same water from the tanks that supplied the home tanks of the experimental fish.

Reviewers' Comments:

Reviewer #1 (Remarks to the Author):

This article suggests that it is the first to demonstrate the emergence of consistent individual differences in behavior (personality) when genetically-identical animals are raised under the same environmental conditions. This is not the case. For instance, for many years, psychologists have attempted to minimize behavioral variability in rodents by using highly inbred (virtually isogenic) strains, and then rearing their subjects under highly controlled, standardized conditions. Even in this situation, consistent individual differences in behavior (personality) emerge over time in the experimental subjects (see Lewejohann et al. 2011, *Dev Psychobiol* 53: 624–630 for a review of this literature). In addition to demonstrating the emergence of personality in isogenic animals reared under the same conditions, these researchers have advanced the same explanations for these patterns as those offered by the current authors in their discussion.

As far as I can tell, the main advance of the current study is that rather than using domesticated strains of rodents, it used a free-living, clonal species (Amazon mollies). Another advance over the rodent studies was that the fish were only allowed limited (olfactory) contact with conspecifics during development, whereas the rodents were reared in small groups. In both cases, the rationale for allowing subjects access to stimuli from conspecifics was that individuals developed behavioral abnormalities if they were reared in strict isolation. However, the fish clearly had less opportunity to engage in social interactions with conspecifics while developing than was the case for the rodents.

Generally speaking, the authors have done a good job in controlling for at least some of the factors that others have shown can affect behavioral development (e.g. identity and age of the mother, prior experiences of the mother, clutch size). While it is clearly impossible to control experimentally or statistically for ALL of the factors that might lead to individual differences in development, with the exception of individual differences in epigenetic factors (see below) the authors did a reasonable job of controlling for many of them.

The paper is generally well-written (although there is some repetition) and the experimental design and statistical analyses seem appropriate to the task at hand.

Other comments

1. Lines 129-133: "Most current explanations of behavioural individuality are based on either of three factors (or an interaction of those): (i) inherent (i.e. genetic and epi-genetic) differences between individuals, (ii) between-individual differences in 'major' ecological factors like

mortality risk, predation regime, resource abundances and competitive regime and/or (iii) direct social interactions. Our findings (0-day treatment) show that behavioural individuality emerges even in settings that are designed to eliminate these factors.

I have two issues with this statement (the gist of which is repeated several times in the article (e.g. see lines 8-15, lines 162-167))

a) The authors neither measured nor controlled for epigenetic differences among the individuals in their experiment. Studies of other species of clonal fish have revealed high levels of both stochastic and environmentally-induced variation in epigenetic factors for individuals with the same genotype (for a recent example, see Leung et al. 2016, *Ecol & Evol.* 2016). Other authors have suggested that high levels of stochastic epigenetic variation might be adaptive in clonal organisms, because they provide a way for parents to produce phenotypically diverse offspring, in spite of their genetic homogeneity ("bet-hedging"). And there is now a sizeable literature indicating that for organisms with the same genotype, epigenetic variants, either stochastic or environmentally induced, can have major impacts on phenotypic development. In any case, it is not true that the authors 'eliminated' epigenetic variation as a factor that might have contributed to the individual differences in behavior observed in the current study.

b) The statement cited above provides a reasonable summary of recent theoretical models of the EVOLUTION of personality, but the authors might also want to consult recent theoretical models on the DEVELOPMENT of personality (which is, of course, the topic of this article). For instance, the authors might want to take a look at Sih et al. 2015 (*TREE*, vol 30). They show how models based on positive feedback loops between internal state and behavior can encourage the emergence of consistent individual differences in behavior over ontogeny, even in the absence of any genetic differences or differences among individuals in environmental factors such as food abundance, presence of predators, social interactions, etc.. The positive-feedback models in Sih et al imply that even if individuals started out with nearly-identical phenotypes, minor (perhaps stochastic) differences among them in initial condition or state, coupled with positive feedbacks between internal state and behavior, could encourage the emergence of personality over time. Some positive feedback loops can occur even if isogenetic subjects are reared under the same strictly-controlled external environmental conditions... for instance, slightly higher insulin levels encourage higher feeding rates, which lead to physiological changes which further increase insulin levels, which encourage even higher feeding rates..... etc. etc.

Reviewer #2 (Remarks to the Author):

I think the authors have done a wonderful job clearly responding to my queries. I also think that they've done a wonderful job responding to the other reviewers' queries. I was unable to see the supplementary material in this submission. However, with the additional analyses, clarity in placing their results in better context, and other editorial changes, I believe the paper is vastly improved.

Dan Blumstein

Reviewer #3 (Remarks to the Author):

I have been asked to review this paper a second time after receiving response from the authors and their revised manuscript. To avoid repeating all details of my previous review I will concentrate on the main issues below, and give some further comments on the authors' response.

One of my concerns was that although the experimental design was good in many ways, the authors have not minimized environmental influences in all respects.

First, I addressed the fact that individuals were not chemically isolated, especially since this species has been shown to use olfactory communication to recognize conspecifics (Reding & Cummings, 2016 still not cited in the revised manuscript).

The authors motivate their choice of design in the following way: "We chose to allow the individuals to have access to olfactory cues to ensure a sufficiently high level of "normalcy" in their development. These animals do not occur in isolation in the wild and the removal of all cues might easily result in abnormal behavior (as shown for several fish species when reared in total isolation)".

This is a valid point, also for ethical reasons. Individuals of some fish species do experience stress in isolation - but if behavior is assumed to be abnormal in total isolation, how should we then interpret individual behavior when fish only have olfactory contact with conspecifics? - partly normal and partly resulting from stress? - but not as much as in total isolation? On the one hand the authors seem to be arguing that the chemical environment is not an important environmental factor (Reding and Cummings work actually tend to suggest the opposite), while simultaneously arguing that it is a prerequisite for normal behavior? I fully understand the logistic and resource limitations of lab-experiments, but it is hard to interpret the results here without an additional treatment with total isolation.

Second, I criticized the fact that two types of food were used of which one was live prey which will potentially increase the scope for individual prey specialization compared to if one prey type was used.

The authors response: "Finally, with regards to two types of food used, this was done because we wanted to provide all fish ad libitum with the best possible diet. The goal of our design was to (a) reduce size (growth) differences due to competition for food in groups (in fact, our treatment fish did not differ in size) and (b) help to maintain good health of all fish throughout the experiment on the long run (which is often difficult with a single food source)."

Far from being an expert on the feeding requirements of this particular species – but considering feeding experiments on many other fish species - I am not convinced that the goals stated above could not have been achieved by providing the experimental fish with one type of high-quality food in sufficient amounts (ad lib).

My third main comment was a questioning of the authors' claim that they are challenging a general paradigm suggesting that sociality inevitably leads to increased individuality. In contrast, for me the results are actually not that unexpected given the strong sociality of the species studied, which might promote conformity in behaviour, and the fact that Edenbrow and Croft on obtained qualitatively similar result in their previous work on killifish.

Some recent reviews/theories suggest that sociality can promote individual differentiation in behaviour and I definitely agree that such mechanisms (discussed in more detail in the previous review) can lead to increased individual variation in some cases - but I would still argue that these effects should depend critically on the ecology of the species studied. As mentioned previously, there are alternative theoretical work and empirical studies suggesting that social learning can lead to conformity, especially under conditions when individual information is unreliable (papers by e.g. Laland, Brown, Giraldeau, van Bergen and others). Indeed, the species here forms large female groups in the wild which potentially could function as information centers and thus lead to conformity in behavior. The authors have incorporated this alternative possibility to some extent in the revised manuscript.

The authors discuss the consequences of formation of dominance hierarchies for adjustment of individual behavior:

"More specifically, our study species of fish generally form dominance hierarchies in the wild which requires that individuals adjust their behavior to avoid conflict and competition, a mechanism likely to generate individual differences. Finally, there is some empirical evidence to suggest that increasing time spent within social groups does drives the emergence of behavioral individuality (Laskowski & Pruitt ProcB, ref no. 41)."

However, there is an important discrepancy between the field observations on large groups of individuals and the lab studies which provide the empirical data. Although the lab studies are interesting and well conducted they are all (as far as I could find out) conducted in the lab using only 2-3 females together. In contrast, in the wild the fish are reported to occur in large schools which, according to resource defense theory (and extensive empirical testing of this theory), should reduce aggressive and territorial behavior considerably because the cost of aggressive and territorial behavior increases with competitor density. Therefore, the intensity of agonistic social interactions in large schools in the wild may be exaggerated when interpreted from lab-studies involving only 2-3 individuals.

In summary, I do think this is a generally well conducted study addressing an interesting problem, and that it deserves to be published. Moreover, the paper has been further improved by the revision. However, with all respect for the authors' work, I still do not think that the results are quite novel and convincing enough to warrant publication in Nature Communications.

Reviewers'; comments:

Reviewer #1 (Remarks to the Author):

Thank you for your very positive response and your highly constructive comments on our revision. Closely following your comments and suggestions, we have taken substantial efforts to further improve our manuscript and we are confident that we have addressed all comments below.

This article suggests that it is the first to demonstrate the emergence of consistent individual differences in behavior (personality) when genetically-identical animals are raised under the same environmental conditions. This is not the case. For instance, for many years, psychologists have attempted to minimize behavioral variability in rodents by using highly inbred (virtually isogenic) strains, and then rearing their subjects under highly controlled, standardized conditions. Even in this situation, consistent individual differences in behavior (personality) emerge over time in the experimental subjects (see Lewejohann et al. 2011, *Dev Psychobiol* 53: 624–630 for a review of this literature). In addition to demonstrating the emergence of personality in isogenic animals reared under the same conditions, these researchers have advanced the same explanations for these patterns as those offered by the current authors in their discussion.

As far as I can tell, the main advance of the current study is that rather than using domesticated strains of rodents, it used a free-living, clonal species (Amazon mollies). Another advance over the rodent studies was that the fish were only allowed limited (olfactory) contact with conspecifics during development, whereas the rodents were reared in small groups. In both cases, the rationale for allowing subjects access to stimuli from conspecifics was that individuals developed behavioral abnormalities if they were reared in strict isolation. However, the fish clearly had less opportunity to engage in social interactions with conspecifics while developing than was the case for the rodents.

In response to the above comment, we have now completely reworked both the introduction and the discussion of our manuscript in order to (i) better acknowledge previous work with genetically identical individuals raised under the same environmental conditions and (ii) discuss the main advance of our study compared to those studies.

To be more specific, in our newly added introductory paragraph (lines 36ff), we now write:

“Previous research has shown that substantial between-individual variation in morphological and physiological traits still develops even among genetically identical individuals reared under highly standardized conditions⁹⁻¹¹. This suggests that – even in the absence of genetic and environmental differences – maternal and epigenetic effects and/or minute experiential/environmental differences during development act as important drivers underlying phenotypic variation¹²⁻¹⁴. Up to now, only a handful of studies have investigated whether these same mechanisms can promote behavioural individuality in the absence of genetic and environmental differences between individuals^{10,15-17}. Most prominently, recent studies on highly inbred mice find that behavioural individuality emerges among genetically identical

individuals^{18,19} when reared in the same environment. The interpretation of these findings, however, is hampered by the fact that individuals were (necessarily) reared in groups. Direct social interactions among conspecifics are well-known to be a powerful factor affecting the development of individuality. In particular, dependent on the species ecology and other factors, direct social interactions may either promote the development of individuality (e.g. via the formation of social interaction structures like social hierarchies; via processes like frequency dependence, niche- and role- specialization^{7, 20-24}) or inhibit the development of individuality (e.g. via positive frequency-dependent social learning and benefits of conformity²⁵⁻²⁷). At present, it is thus not clear (i) whether and to what extent individuality emerges among genetically identical individuals reared in isolation under highly standardized environmental conditions and (ii) how a change in specific ecological factors (here: social rearing conditions) affects the development of individuality, compared to this null model scenario. The goal of this study was to investigate these two central issues about the causes of behavioural individuality."

To give a second example, in order to achieve the above goals, we have now reworked our reference list to include key references on genetically identical individuals (i.e.: Gärtner 1990, Archer 2003, Vogt 2008, Schütt 2011, Lewejohann 2011, Croft 2013, Freund 2013).

Generally speaking, the authors have done a good job in controlling for at least some of the factors that others have shown can affect behavioral development (e.g. identity and age of the mother, prior experiences of the mother, clutch size). While it is clearly impossible to control experimentally or statistically for ALL of the factors that might lead to individual differences in development, with the exception of individual differences in epigenetic factors (see below) the authors did a reasonable job of controlling for many of them.

In response to this comment and the associated point 1a below, both in the introduction and the discussion of our manuscript we now clearly acknowledge that we did not control for epigenetic differences between individuals and we now discuss in detail how epigenetic differences between individuals provide one promising candidate explanation for our findings (please see our detailed response to point 1a below).

The paper is generally well-written (although there is some repetition) and the experimental design and statistical analyses seem appropriate to the task at hand.

Other comments

1. Lines 129-133: "Most current explanations of behavioural individuality are based on either of three factors (or an interaction of those): (i) inherent (i.e. genetic and epi-genetic) differences between individuals, (ii) between-individual differences in 'major' ecological factors like mortality risk, predation regime, resource abundances and competitive regime and/or (iii) direct social interactions. Our findings (0-day treatment) show that behavioural individuality emerges even in settings that are designed to eliminate these factors.

I have two issues with this statement (the gist of which is repeated several times in the article (e.g. see lines 8-15, lines 162-167))

a) The authors neither measured nor controlled for epigenetic differences among the individuals in their experiment. Studies of other species of clonal fish have revealed high levels of both stochastic and environmentally-induced variation in epigenetic factors for individuals with the same genotype (for a recent example, see Leung et al. 2016, *Ecol & Evol.* 2016). Other authors have suggested that high levels of stochastic epigenetic variation might be adaptive in clonal organisms, because they provide a way for parents to produce phenotypically diverse offspring, in spite of their genetic homogeneity ("bet-hedging"). And there is now a sizeable literature indicating that for organisms with the same genotype, epigenetic variants, either stochastic or environmentally induced, can have major impacts on phenotypic development. In any case, it is not true that the authors 'eliminated' epigenetic variation as a factor that might have contributed to the individual differences in behavior observed in the current study.

In response to this comment, we now clearly acknowledge that we did not control for epigenetic differences between individuals and – closely following this and the following point (1b) of this referee – we now discuss in detail how

- (i) epigenetic differences between individuals (this point) and/or
- (ii) minute experiential differences in combinations with positive feedbacks (point 1b by the same referee – please see below for details how we addressed this point)

are two promising candidate explanations underlying our experimental findings and the emergence of individuality more generally.

Most importantly, in order to achieve this goal, we have now completely reworked our discussion. To be specific, we now discuss the importance of epigenetic differences in a completely new paragraph (lines 122ff):

“Importantly, our finding puts emphasis on two alternative factors.

[It follows a discussion of minute experiential variation between individuals and positive feedbacks, addressing point 1b of the referee, see below.]

(lines 152ff): *“Second, epigenetic variation among individuals, which may be either stochastic or environmentally induced. While we took substantial efforts to limit the influence of this sort of variation (e.g. through highly standardized environmental conditions, our split brood design, using mothers of same age/experience), it is experimentally impossible to fully eliminate epigenetic variation among individuals and the observed individuality in our experiment may be driven by this factor¹²⁻¹⁴. Whether and under which conditions such between-individual epigenetic variation (and thus the observed individuality) is the result of an adaptive strategy is*

currently an open question. Mothers may, for example, employ a bet-hedging strategy to generate adaptive epigenetic variation among her offspring^{12,14}; this may be particularly beneficial in unpredictably changing environments or in clonal organisms where standing genetic variation is lacking. Furthermore between-individual epigenetic variation may be an adaptive response to minute experiential/environmental differences between individuals (see above), thereby potentially contributing to a positive feedback loop reinforcing such differences^{11,14}. Alternatively, between-individual epigenetic variation may not be the result of natural selection, but rather our experiment (and others like it) may have captured the natural stochasticity that occurs during the developmental process^{11,39-43}.”

Moreover, we now highlight the importance of epigenetic differences (and minute experiential variation in combination with positive feedbacks – point 1b of the referee below) already in the introduction of our manuscript (lines 36ff):

“Previous research has shown that substantial between-individual variation in morphological and physiological traits still develops even among genetically identical individuals reared under highly standardized conditions⁹⁻¹¹. This suggests that – even in the absence of genetic and environmental differences – maternal and epigenetic effects and/or minute experiential/environmental differences during development act as important drivers underlying phenotypic variation¹²⁻¹⁴.”

Furthermore, throughout our manuscript, we now stress that most research on adaptive individuality focusses on differences in genes and/or environmental conditions and that this is in contrast to our findings which suggest that epigenetic differences or minute experiential variation in combination with positive feedbacks (point (1b) of the referee below) provide a likely explanation. For example (lines 118ff):

“Here we report experimental evidence that substantial behavioural individuality emerges even among genetically identical individuals housed under “identical” (i.e. highly standardized) environmental conditions. This finding is in contrast to the current research paradigm associated with adaptive individuality which focusses on differences in genes and/or environmental conditions (including the social environment). Importantly, our finding puts emphasis on two alternative factors.”

Finally, and associated with the above revisions, we have now included several key and up to date references to epigenetic variation between individuals and its potential importance for individuality (i.e. Gärtner 1990, Wong 2005, Lewejohann 2011, Vogt 2015, Leung 2016).

b) The statement cited above provides a reasonable summary of recent theoretical models of the EVOLUTION of personality, but the authors might also want to consult recent theoretical models on the DEVELOPMENT of personality (which is, of course, the topic of this article). For instance, the authors might want to take a look at Sih et. al. 2015 (TREE, vol 30). They show how models based on positive feedback loops between internal state and behavior can encourage the emergence of consistent individual differences in behavior over ontogeny, even in the absence of any genetic

differences or differences among individuals in environmental factors such as food abundance, presence of predators, social interactions, etc.. The positive-feedback models in Sih et al imply that even if individuals started out with nearly-identical phenotypes, minor (perhaps stochastic) differences among them in initial condition or state, coupled with positive feedbacks between internal state and behavior, could encourage the emergence of personality over time. Some positive feedback loops can occur even if isogenetic subjects are reared under the same strictly-controlled external environmental conditions... for instance, slightly higher insulin levels encourage higher feeding rates, which lead to physiological changes which further increase insulin levels, which encourage even higher feeding rates..... etc. etc.

As mentioned above, in response to this and the above point 1a, we have now completely reworked the introduction and discussion of our manuscript to acknowledge and discuss in detail how

- (i) epigenetic differences between individuals (point 1a above) and/or
- (ii) minute experiential differences in combinations with positive feedbacks (this point)

are two promising candidate explanations underlying our experimental findings and the emergence of individuality more generally. As mentioned above, in order to achieve this goal, we have completely reworked our discussion. To be specific, we now extensively discuss the importance of minute experiential variation and positive feedback loops in a completely new paragraph (lines 122ff):

“Importantly, our finding puts emphasis on two alternative factors.

First, minute and stochastic experiential/environmental variation between individuals. In nature, no two individuals experience identical environmental conditions over development; similarly, experimentally, it is practically impossible to provide identical experiential/environmental conditions to different individuals over development. Thus, despite our best efforts, it is likely that different individuals experienced different micro-environments such as slight differences in water temperature, olfactory signals or distribution of prey items. Recent theory on the adaptive development of individuality suggests that even among initially (genetically and epi-genetically) identical individuals, such minute environmental or experiential differences can induce positive feedback loops that eventually set individuals down to different developmental trajectories³³⁻³⁶. For example, in order to (i) ensure a sufficiently high level of normalcy in development and (ii) make the 0-day treatment act as a proper control for our social treatments by explicitly controlling for direct social interactions per se, we allowed our isolated individuals (0-day treatment) to have access to olfactory cues of conspecifics (see Methods). As Amazon mollies are known to use olfactory cues to detect conspecifics³⁷ and lineage-kin³⁸, it is conceivable that stochastic variation in such chemical cues between individuals (in combination with positive feedbacks) may be a factor driving the development of individuality in our experiments. To give a second example, in order to (i) avoid size (growth) differences between individuals due to

competition for food and (ii) maintain good health of all fish throughout the experiment, we provided our individuals with two types of food sources (see Methods). Again, it is conceivable that stochastic variation in the distribution and movement of the food and/or developing food preferences for either source (in combination with positive feedbacks) may be a factor driving the development of individuality in our experiments. Future work that closely follows the developmental experiences and the associated behavioural responses of individuals should aim to elucidate when and how such minute experiential differences can trigger the development of individuality. Moreover, an especially compelling question is whether developmental divergence triggered by such minute experiential differences makes the emergence and patterning of individuality inherently unpredictable.

Second,...” [it follows a discussion on epigenetic differences, addressing point (1a) of the referee, see above].

Moreover, as discussed above, we now highlight the importance minute experiential variation in combination with positive feedbacks already in the introduction of our manuscript (for details, see our response to point 1(a) above), (ii) throughout our manuscript, we now stress that most research on adaptive individuality focusses on differences in genes and environmental conditions and that this is in contrast to our findings which suggests that epigenetic differences or minute experiential variation in combination with positive feedbacks (point (1b) of the referee below) provide a likely explanation (for details, see our response to point 1(a) above).

Finally, associated with these revisions, we have now included several key and up to date references to minute experiential differences and positive feedbacks and its potential importance for individuality (i.e.: Luttbeg & Sih 2010, Dingemanse & Wolf 2010, Frankenhuis 2011, Sih 2015).

Reviewer #2 (Remarks to the Author):

I think the authors have done a wonderful job clearly responding to my queries. I also think that they've done a wonderful job responding to the other reviewers' queries. I was unable to see the supplementary material in this submission. However, with the additional analyses, clarity in placing their results in better context, and other editorial changes, I believe the paper is vastly improved.

Dan Blumstein

Many thanks again for your previous comments which allowed us vastly improve our manuscript.

Reviewer #3 (Remarks to the Author):

I have been asked to review this paper a second time after receiving response from the authors and their revised manuscript. To avoid repeating all details of my previous review I will concentrate on the main issues below, and give some further comments on the authors' response.

Many thanks for taking the time to review our paper a second time. As during our first revision, we highly appreciate your constructive comments and – based on your comments (also going back to the comments of your previous report) – we have taken substantial efforts to further improve our manuscript.

One of my concerns was that although the experimental design was good in many ways, the authors have not minimized environmental influences in all respects.

First, I addressed the fact that individuals were not chemically isolated, especially since this species has been shown to use olfactory communication to recognize conspecifics (Reding & Cummings, 2016 still not cited in the revised manuscript).

While we address the broader issue in detail below, we here note that we have now included this important reference into our manuscript (lines 137ff):

“As Amazon mollies are known to use olfactory cues to detect conspecifics³⁷ [reference to Reding and Cummings 2016] and lineage-kin³⁸, [reference to Makowicz et al. 2016] it is conceivable that stochastic variation in such chemical cues between individuals (in combination with positive feedbacks) may be a factor driving the development of individuality in our experiments.”

The authors motivate their choice of design in the following way: "We chose to allow the individuals to have access to olfactory cues to ensure a sufficiently high level of “normalcy” in their development. These animals do not occur in isolation in the wild and the removal of all cues might easily result in abnormal behavior (as shown for several fish species when reared in total isolation)".

This is a valid point, also for ethical reasons. Individuals of some fish species do experience stress in isolation - but if behavior is assumed to be abnormal in total isolation, how should we then interpret individual behavior when fish only have olfactory contact with conspecifics? - partly normal and partly resulting from stress? - but not as much as in total isolation? On the one hand the authors seem to be arguing that the chemical environment is not an important environmental factor (Reding and Cummings work actually tend to suggest the opposite), while simultaneously arguing that it is a prerequisite for normal behavior? I fully understand the logistic and resource limitations of lab-experiments, but it is hard to interpret the results here without an additional treatment with total isolation.

As detailed in our previous response (and in light of our previous experience with our study system), we maintain that olfactory cues are a necessary and sufficient conditions to ensure a sufficiently high

level of normalcy in the development of our experimental individuals. We note that our findings strongly support this statement as the behavior of all the fish did not differ between our social and our non-social treatments; strongly suggesting that exposure to chemical cues is sufficient to support the development of “normalcy” in behavior.

More generally, in the comment above, and the related comments in the previous report by the same referee, the referee raises the important point that minute and non-controlled variation in olfactory cues (this point) and – relatedly – the fact that our individuals were fed with two types of food sources (following point by the referee) could have differentially affected the behavioral development of our individuals and thus potentially explain the emergence of individuality in our experiments.

While we fully agree with the referee, we stress that this does not weaken the strength of our conclusion: a key finding of our study is that individuality emerges among genetically identical individuals isolated directly after birth into highly standardized environmental conditions. This finding contrasts the current research paradigm associated with adaptive individuality which focusses on differences in genes and/or environmental conditions (including the social environment) and puts emphasis on epigenetic variation and/or minute experiential variation between individuals as important drivers underlying individuality.

We thus fully agree with the referee that (next to epigenetic variation, see above) minute experiential variation in olfactory cues (this point) and the fact that two different types of food sources were used (following point by the referee) are two potential candidates triggering minute experiential variation between individuals. We have now taken substantial effort to acknowledge and discuss this possibility in our manuscript.

Most importantly, we have now completely reworked our discussion in order to (i) discuss the importance of minute experiential variation and positive feedbacks in general and (ii) closely following the comments of the referee, discuss in detail how minute variation in chemical cues (this point) and the fact that individuals in our experiments had access to two food sources (following point by the referee) can explain our results (lines 122ff):

“Importantly, our finding puts emphasis on two alternative factors.

First, minute and stochastic experiential/environmental variation between individuals. In nature, no two individuals experience identical environmental conditions over development; similarly, experimentally, it is practically impossible to provide identical experiential/environmental conditions to different individuals over development. Thus, despite our best efforts, it is likely that different individuals experienced different micro-environments such as slight differences in water temperature, olfactory signals or distribution of prey items. Recent theory on the adaptive development of individuality suggests that even among initially (genetically and epi-genetically) identical individuals, such minute environmental or experiential differences can induce positive feedback loops that eventually set individuals down to different developmental trajectories³³⁻³⁶. For example, in order

to (i) ensure a sufficiently high level of normalcy in development and (ii) make the 0-day treatment act as a proper control for our social treatments by explicitly controlling for direct social interactions per se, we allowed our isolated individuals (0-day treatment) to have access to olfactory cues of conspecifics (see Methods). As Amazon mollies are known to use olfactory cues to detect conspecifics³⁷ and lineage-kin³⁸, it is conceivable that stochastic variation in such chemical cues between individuals (in combination with positive feedbacks) may be a factor driving the development of individuality in our experiments. To give a second example, in order to (i) avoid size (growth) differences between individuals due to competition for food and (ii) maintain good health of all fish throughout the experiment, we provided our individuals with two types of food sources (see Methods). Again, it is conceivable that stochastic variation in the distribution and movement of the food and/or developing food preferences for either source (in combination with positive feedbacks) may be a factor driving the development of individuality in our experiments. Future work that closely follows the developmental experiences and the associated behavioural responses of individuals should aim to elucidate when and how such minute experiential differences can trigger the development of individuality. Moreover, an especially compelling question is whether developmental divergence triggered by such minute experiential differences makes the emergence and patterning of individuality inherently unpredictable.

Second,... [it follows a discussion on epigenetic differences as an explanatory candidate for our findings].”

Also, we have now reworked our introduction to emphasize that minute experiential variation may be an important driver underlying the development of individuality (line 36):

“Previous research has shown that substantial between-individual variation in morphological and physiological traits still develops even among genetically identical individuals reared under highly standardized conditions⁹⁻¹¹. This suggests that – even in the absence of genetic and environmental differences – maternal and epigenetic effects and/or minute experiential/environmental differences during development act as important drivers underlying phenotypic variation¹²⁻¹⁴.”

Moreover, associated with these revisions, we have now included several recent and key references in our paper that underscore the potential importance of minute experiential variation between individuals for the development of individuality (i.e.: Luttbegg & Sih 2010, Dingemanse & Wolf 2010, Frankenhuis 2011, Sih 2015).

To give a last example, throughout the manuscript, we have carefully reworked our wording in order to avoid the impression that our individuals experienced completely identical environments. For example, in the discussion we now write (lines 125ff):

“In nature, no two individuals experience identical environmental conditions over development; similarly, experimentally, it is practically impossible to provide identical experiential/environmental conditions to different individuals over development. Thus, despite our best efforts, it is likely that different individuals experienced different micro-environments such as slight differences in water temperature, olfactory signals or distribution of prey items.”

In line with this, throughout the manuscript, we have now decided to use the more appropriate description of “highly standardized environmental conditions” instead of “identical environment” (see, for example, the abstract or the first paragraph of our introduction).

Second, I criticized the fact that two types of food were used of which one was live prey which will potentially increase the scope for individual prey specialization compared to if one prey type was used.

The authors response: "Finally, with regards to two types of food used, this was done because we wanted to provide all fish ad libitum with the best possible diet. The goal of our design was to (a) reduce size (growth) differences due to competition for food in groups (in fact, our treatment fish did not differ in size) and (b) help to maintain good health of all fish throughout the experiment on the long run (which is often difficult with a single food source)."

Far from being an expert on the feeding requirements of this particular species – but considering feeding experiments on many other fish species - I am not convinced that the goals stated above could not have been achieved by providing the experimental fish with one type of high-quality food in sufficient amounts (ad lib).

As detailed in our previous response (and in light of our previous experience with our study system), we maintain that two types of food sources were necessary in order to both (a) reduce size (growth) differences due to competition for food in groups and (b) maintain good health of all fish. To be more specific on the latter point, newly born juveniles in our study system consistently experience substantial mortality rates if not given live food (Artemia) – they just don't like to come up to the surface to eat flake food as small juveniles. Additionally, as the fish get older and larger, it becomes increasingly difficult to offer them enough Artemia to satiate them without leading to a build-up of dead Artemia in the tanks (Artemia are salt-water animals, and after adding them to the freshwater, they quickly die). Continually adding more and more Artemia thus leads to a build-up of dead Artemia in the aquariums which is seriously detrimental to fish health. We also note that both the absence of size differences between individuals and the very low mortality rate (only two fish died over the course of the experiment) provide a further justification for our experimental design.

That said, and as discussed in detail above, we fully agree with the referee that it is easily conceivable that the presence of two types of food sources (this point) and/or olfactory cues (previous point) could have differentially affected the behavioural development of our individuals and thus potentially explain the emergence of individuality in our experiments. While this does weaken the strength of our conclusions (see our discussion above), we have now taken substantial efforts to

discuss this important possibility in our manuscript – please see our detailed discussion above. To briefly summarize again, we have now substantially reworked our discussion in order to (i) discuss the importance of minute experiential variation and positive feedbacks in general and (ii) closely following the comments and suggestions of the referee, discuss in detail how the presence of two food sources (this point) and/or chemical cues (previous point) can explain our results. Moreover, we have (i) reworked our introduction to emphasize that already that minute experiential variation may be an important driver underlying the development of personality differences (ii) associated with these revisions, we have now included several recent and key references in our paper that underscore the potential importance of slight experiential variation between individuals for the development of individuality (see above) and (iii) we have carefully reworked manuscript our manuscript in order to avoid the impression that our individuals experienced completely identical environments. Please see above for details for each of these points.

My third main comment was a questioning of the authors' claim that they are challenging a general paradigm suggesting that sociality inevitably leads to increased individuality. In contrast, for me the results are actually not that unexpected given the strong sociality of the species studied, which might promote conformity in behaviour, and the fact that Edenbrow and Croft obtained qualitatively similar result in their previous work on killifish.

Some recent reviews/theories suggest that sociality can promote individual differentiation in behaviour and I definitely agree that such mechanisms (discussed in more detail in the previous review) can lead to increased individual variation in some cases - but I would still argue that these effects should depend critically on the ecology of the species studied. As mentioned previously, there are alternative theoretical work and empirical studies suggesting that social learning can lead to conformity, especially under conditions when individual information is unreliable (papers by e.g. Laland, Brown, Giraldeau, van Bergen and others). Indeed, the species here forms large female groups in the wild which potentially could function as information centers and thus lead to conformity in behavior. The authors have incorporated this alternative possibility to some extent in the revised manuscript.

The authors discuss the consequences of formation of dominance hierarchies for adjustment of individual behavior:

"More specifically, our study species of fish generally form dominance hierarchies in the wild which requires that individuals adjust their behavior to avoid conflict and competition, a mechanism likely to generate individual differences. Finally, there is some empirical evidence to suggest that increasing time spent within social groups does drives the emergence of behavioral individuality (Laskowski & Pruitt ProcB, ref no. 41)."

However, there is an important discrepancy between the field observations on large groups of individuals and the lab studies which provide the empirical data. Although the lab studies are interesting and well conducted they are all (as far as I could find out) conducted in the lab using only 2-3 females together. In contrast, in the wild the fish are reported to occur in large schools which,

according to resource defense theory (and extensive empirical testing of this theory), should reduce aggressive and territorial behavior considerably because the cost of aggressive and territorial behavior increases with competitor density. Therefore, the intensity of agonistic social interactions in large schools in the wild may be exaggerated when interpreted from lab-studies involving only 2-3 individuals.

We fully agree with the referee that – dependent on the species ecology and other specifics of the situation, direct social interactions may either promote or inhibit the development of individuality. In response to this comment (and the related comments by the same referee in the previous report) we have now substantially reworked both the introduction and the discussion of our manuscript in order to present a balanced view on this issue (i.e. present both predictions as possible outcomes of our experiment). To be concrete, in a completely new written paragraph in the introduction we now write (lines 47ff):

“Direct social interactions among conspecifics are well-known to be a powerful factor affecting the development of individuality. In particular, dependent on the species ecology and other factors, direct social interactions may either promote the development of individuality (e.g. via the formation of social interaction structures like social hierarchies; via processes like frequency dependence, niche- and role-specialization^{7, 20-24}) or inhibit the development of individuality (e.g. via positive frequency-dependent social learning and benefits of conformity²⁵⁻²⁷). At present, it is thus not clear (i) whether and to what extent individuality emerges among genetically identical individuals reared in isolation under highly standardized environmental conditions and (ii) how a change in specific ecological factors (here: social rearing conditions) affects the development of individuality, compared to this null model scenario. The goal of this study was to investigate these two central issues about the causes of behavioural individuality.”

Also, in response to the above issue, and in order to present a balanced view on the consequences of direct social interactions for the development of individuality, we have now also drafted a completely new paragraph in the discussion of our paper (lines 168ff):

“It is currently thought that direct social interactions are a powerful causal factor affecting the development of individuality. As discussed above, dependent on the species ecology and other factors such as density, risk, etc., direct social interactions can be predicted to either promote (e.g. via the formation social interaction structures like social hierarchies; via processes like frequency dependence, niche- and role- specialization^{7, 20-24}) or inhibit (e.g. via positive frequency-dependent social learning and benefits of conformity²⁵⁻²⁷) the development of individuality. Up to now, however, few studies¹⁶ have evaluated the importance of direct social interactions with a controlled experimental approach that compares a treatment in which genetically identical individuals are allowed to directly interact with each other with an appropriate null model treatment that does not allow for such interactions. When comparing our 0-day null model

treatment with our 7-day and 28-day direct social experience treatments, we find that the amount of behavioural individuality observed is not affected by the level of direct social experience of individuals. We note that – while we do not find statistically significant differences between treatments – the amount of among- and within-individual variation does tend to decrease in the most social treatment (28-day). This finding is in line with previous studies on clonal fish which did not find an effect of direct social experience on the repeatability of behaviour¹⁶. We do not claim that social processes play no role in the development of behavioural individuality, but our findings strongly suggest that the non-social processes discussed above (i.e., minute experiential differences triggering positive feedback loops; epigenetic variation) may be substantially more important for the development of individuality than currently thought.”

Moreover, we have also rewritten the abstract of our manuscript to follow the suggestions of the referee that sociality might either promote or inhibit individuality (line 12ff):

“Moreover, even among genetically identical individuals, direct social interactions are thought to be a powerful factor shaping the development of individuality.”

Finally, we have included three key references for how social processes can inhibit the development of individuality (Pike & Laland 2010; Webster and Ward 2011; Morgan & Laland 2012).

In summary, I do think this is a generally well conducted study addressing an interesting problem, and that it deserves to be published. Moreover, the paper has been further improved by the revision. However, with all respect for the authors’ work, I still do not think that the results are quite novel and convincing enough to warrant publication in Nature Communications.

We again thank the referee for her/his thoughtful and constructive comments which (in our view) helped us to substantially improve our manuscript.